# Unbiased Watermark for Large Language Models

**Zhengmian Hu**[1], **Lichang Chen**[1], **Xidong Wu**[2], **Yihan Wu**[1], **Hongyang Zhang**[3], **Heng Huang**[1]

[1]Department of Computer Science, University of Maryland, College Park, MD 20742, USA
[2]Department of ECE, University of Pittsburgh, Pittsburgh, PA 15261, USA
[3]School of Computer Science, University of Waterloo, Waterloo, ON N2L 3G1, Canada
`huzhengmian@gmail.com,boblichangchen@gmail.com,xidong_wu@outlook.com,`
`ywu42@umd.edu,hongyang.zhang@uwaterloo.ca,henghuanghh@gmail.com`

## Abstract

The recent advancements in large language models (LLMs) have sparked a growing apprehension regarding the potential misuse. One approach to mitigating this risk is to incorporate watermarking techniques into LLMs, allowing for the tracking and attribution of model outputs. This study examines a crucial aspect of watermarking: how significantly watermarks impact the quality of model-generated outputs. Previous studies have suggested a trade-off between watermark strength and output quality. However, our research demonstrates that it is possible to integrate watermarks without affecting the output probability distribution with appropriate implementation. We refer to this type of watermark as an **unbiased watermark**. This has significant implications for the use of LLMs, as it becomes impossible for users to discern whether a service provider has incorporated watermarks or not. Furthermore, the presence of watermarks does not compromise the performance of the model in downstream tasks, ensuring that the overall utility of the language model is preserved. Our findings contribute to the ongoing discussion around responsible AI development, suggesting that unbiased watermarks can serve as an effective means of tracking and attributing model outputs without sacrificing output quality.

## 1 Introduction

In recent years, large language models (LLMs) (Google, 2023; OpenAI, 2023a;b) have become an indispensable tool for a wide range of tasks, including text generation (Iyer et al., 2022; Chung et al., 2022), translation (Bojar et al., 2017; Barrault et al., 2019), summarization (Liu & Lapata, 2019), etc. With the escalating misuse of LLMs, such as plagiarism, tracking the usage of text generated by machines has become increasingly important. One viable method to monitor the usage of LLMs is watermarking (Gu et al., 2022; Kirchenbauer et al., 2023; Venugopal et al., 2011), which embeds imperceptible information within the generated text, thereby allowing for efficient detection and tracking of the model's potential abuse.

Watermarking techniques can serve multiple purposes, such as embedding ownership information within the generated text to protect the intellectual property rights of the model. It can also help mitigate potential harm caused by LLMs by monitoring where the model is being used and whether it is being misused or abused.

A good watermarking method should not adversely affect the normal usage of the language model or degrade the quality of the generated text. However, a prevailing belief holds that there is an inevitable trade-off between the strength of the watermark and the quality of the output text. For instance, recent work by Kirchenbauer et al. (2023) introduced a method that augmented the logits of a randomly selected set of "green" tokens. By tuning the "magnitude of logits adjustment", they demonstrated a trade-off between watermark strength and text quality.

Our primary contribution is to challenge this conventional wisdom. We show that with the right implementation, watermarking can be accomplished without affecting the output quality. We refer to this particular type of watermark as an **unbiased watermark**. We approach the problem of output quality degradation from the perspective of watermark detection. We posit that if the watermark

causes a decline in output quality, there should be a method to guess the presence of the watermark based on the quality. Conversely, if the watermark cannot be detected, it implies that the output quality remains unaffected. Specifically, we provide a proof that with a suitable implementation, watermarking does not affect the output probability distribution. This has significant implications, as users who do not have the private key are unable to discern whether a service provider has applied watermarking to the model. Furthermore, the addition of watermarking does not affect the performance of the generated text in any downstream tasks. **Our main contributions can be summarized as follows:**

- We introduce *unbiased watermark*, an innovative family of watermark methods that guarantee the non-degradation of text quality. In addition, we offer a comprehensive framework that facilitates the design and detection of unbiased watermarks.

- We propose two innovative and practical watermarking techniques known as $\delta$-reweight and $\gamma$-reweight. Through extensive experimentation, we demonstrate that these techniques preserve output quality in machine translation and text summarization tasks.

- We develop an advanced maximin variant of the original log-likelihood ratio test for watermark detection. This novel detection method comes with theoretical guarantees, specifically an upper bound on type I error, thus enhancing the reliability of watermark detection in language models.

## 2 PRELIMINARY

In this section, we delve into the problem of watermarking in the context of LLMs. We begin by setting up the problem and defining essential concepts.

**Problem Modeling:** We first introduce several notations to formalize the problem. Let $\Sigma$ denote the vocabulary set, which is the set of all possible tokens an LLM can generate in a single step. We then define the set $\Sigma^*$ as the collection of all possible strings of any length, including those of length zero.

An LLM generates a sequence of tokens conditioned on a given context. In a single step, the probability of generating the next token $x_{n+1} \in \Sigma$ given the current context, $x_1, x_2, ..., x_n$, can be denoted as $P_M(x_{n+1} \mid x_1, x_2, ..., x_n)$. The LLM operates in an autoregressive fashion, which means the joint probability of generating multiple tokens $x_{n+1}, \ldots, x_{n+m}$ can be written as:

$$P_M(x_{n+1}, \ldots, x_{n+m} \mid x_1, x_2, ..., x_n) = \prod_{i=1}^{m} P_M(x_{n+i} \mid x_1, x_2, ..., x_n, x_{n+1}, \ldots, x_{n+i-1}).$$

For simplicity, we use the following notation: $P_M(\boldsymbol{x}_{n+1:n+m} \mid \boldsymbol{x}_{1:n})$, where $\boldsymbol{x}_{n+1:n+m} = (x_{n+1}, \ldots, x_{n+m}) \in \Sigma^*$.

In the context of watermarking, we introduce a service provider that holds a private key $k$ from the key space $K$. The key $k \in K$ is chosen at random from the prior distribution $P_K(k)$. The watermarked output of the LLM follows distribution $P_{M,w}(x_{n+1} \mid x_1, x_2, ..., x_n; k)$, which is conditioned on both the key $k$ and the context $\boldsymbol{x}_{1:n}$. Similarly, we use the notation $P_{M,w}(\boldsymbol{x}_{n+1:n+m} \mid \boldsymbol{x}_{1:n}; k)$ for the probability of generating a sequence of tokens in a watermarked model.

**Objective.** Our goal is to devise a watermarking scheme that: a) is efficiently detectable by the service provider; b) can't be detected by users and does not negatively impact the quality of the output.

The reason we focus on the detection of watermarks by users is that it is closely related to the output quality. If the watermark causes a degradation in the output quality, there should exist a method to infer the presence of the watermark by examining the quality. Conversely, if the watermark is undetectable, it implies that it does not impact the output quality.

From a statistical testing perspective, a watermark is considered strictly undetectable if the probability distributions of the watermarked and non-watermarked outputs are identical. To capture this notion, we define several desirable properties of watermarking schemes.

**Definition 1** ($n$-shot-undetectable). *For a fixed input sequence $\boldsymbol{a} \in \Sigma^*$, we say that watermarked LLM and key prior pair $(P_{M,w}, P_K)$ is $n$-shot-undetectable compared to original LLM $P_M$ if*

$$\prod_{i=1}^{n} P_M(\boldsymbol{x}^i \mid \boldsymbol{a}) = \sum_{k \in K} P_K(k) \prod_{i=1}^{n} P_{M,w}(\boldsymbol{x}^i \mid \boldsymbol{a}; k), \quad \text{for any } n \text{ number of strings } \boldsymbol{x}^i \in \Sigma^*.$$

**Definition 2** (downstream-invariant). *We say the watermarked LLM and key prior pair $(P_{M,w}, P_K)$ are invariant compared to original LLM $P_M$ on downstream tasks iff*

$$\mathbb{E}_{\boldsymbol{x} \sim P_{M,w}(\cdot|\boldsymbol{a};k),k\sim P_K}[f(\boldsymbol{x})] = \mathbb{E}_{\boldsymbol{x} \sim P_M(\cdot|\boldsymbol{a})}[f(\boldsymbol{x})],$$

*for any strings $\boldsymbol{x}, \boldsymbol{a} \in \Sigma^*$, and for any metric $f : \Sigma^* \to \mathbb{R}$.*

Note that the one-shot-undetectable property implies the downstream invariant property, as identical distributions yield identical expectations for any function. Interestingly, this implication does not require the $n$-shot-undetectable property for $n > 1$, which means a watermarking scheme that is one-shot-undetectable can still maintain the output quality for downstream tasks even if the user might discern the existence of the watermark through multiple generation requests.

In summary, we have outlined the preliminary concepts and objectives for developing a watermarking scheme for LLMs. We highlight the desired properties of $n$-shot-undetectability and downstream invariance, as they provide a rigorous theoretical guarantee of quality preservation and integrity in the deployment of watermark schema. In Section 4, we will present a watermark framework that is provably $n$-shot-undetectable for any given integer $n \geq 1$.

## 3  WARM UP: UNDETECTABILITY IN A SIMPLIFIED TOY ENVIRONMENT

In this subsection, we aim to prove the feasibility of undetectability in a highly simplified toy environment. This preliminary analysis serves as a foundation for understanding the more complex scenarios that follow.

**Settings.** Consider a service provider that offers a random number generation service. The service outputs a uniformly distributed random number in the set $\{0, 1\}$. The clean generation process can be represented as $P_M(x) = 1/2$, $\forall x \in \{0, 1\}$. We assume that the key $k$ belongs to the set $\{0, 1\}$ and is selected with equal probability. With the watermark added, the probability of the new output can be expressed as: $P_{M,w}(x \mid k) = \delta_k(x)$.

Recall that the one-shot-undetectable property can be represented as $P_M(x) = \sum_{k \in K} P_{M,w}(x \mid k)P_K(k)$. Suppose that a user can only make a single request to the service. If the user is unaware of the key, the user will be unable to distinguish whether the received result is watermarked or not. Therefore, in this simplified scenario, the undetectability of the watermark is achieved.

However, there is a considerable gap between this toy example and the practical implementation of watermarking in LLMs. Firstly, the symbol set $\Sigma$ in LLMs is far more complex than the binary set $\{0, 1\}$, and the probability distribution is not uniform. Besides, the generation process in LLMs is autoregressive, which means that more than one symbol are generated iteratively. Furthermore, the toy example does not satisfy the $n$-shot-undetectable property for $n > 1$.

Despite these differences, this simple example provides essential insights that help in understanding the following sections where we address these challenges. The underlying principles of undetectability remain constant, while their application becomes more intricate in a more complex environment.

## 4  WATERMARKING WITH UNBIASED REWEIGHTING

In this section, we build upon the intuition from the previous section and extend the approach to LLMs' generation. The section is structured as follows: Section 4.1 introduces a fundamental mathematical tool for addressing the reweighting problem in general discrete probability distributions. Section 4.2 applies the reweighting technique to LLMs. Section 4.3 presents the final framework.

### 4.1  DISTRIBUTION REWEIGHTING

In its most general form, we consider a random watermark code $E$ and a reweight function $R_E : \Delta_\Sigma \to \Delta_\Sigma$, which depends on the random watermark code $E$. The set of all possible probability distributions on the symbol set $\Sigma$ is denoted as $\Delta_\Sigma$, which forms a simplex.

**Definition 3.** *A **reweighting function** is a tuple $(\mathcal{E}, P_E, R)$ where $\mathcal{E}$ is called the watermark code space, $P_E$ is a probability distribution on space $\mathcal{E}$, and $R$ is a function $R : \mathcal{E} \times \Delta_\Sigma \to \Delta_\Sigma$. For a specific watermark code $E \in \mathcal{E}$, we denote the partially evaluated reweighting function as $R_E : \Delta_\Sigma \to \Delta_\Sigma$.*

**Definition 4.** *Given a random watermark code $E$ and a reweighting function $R_E : \Delta_\Sigma \to \Delta_\Sigma$, we say that R is an **unbiased reweighting function** if and only if for all $P \in \Delta_\Sigma$, $\mathbb{E}_E[R_E(P)] = P$.*

### 4.1.1 EXISTING REWEIGHTING METHODS

Kirchenbauer et al. (2023) essentially comprise two reweighting methods in their work, but neither of them satisfies the unbiased property.

Both methods have $\mathcal{E}$ as the set of mappings $f : \Sigma \to \{\text{red}, \text{green}\}$, such that $f$ maps half of the tokens in $\Sigma$ to 'red' and the other half to 'green', and $P_E$ as a uniform distribution. Therefore, the random watermark code $E$ assigns each symbol to either *red* or *green*. The "Hard Red List" method sets the probability of all red symbols to zero and renormalizes the probabilities of the remaining vocabulary. The second method is "Soft Red List" blocking, where they randomly select the same "Red List" as the first method and decrease the corresponding probability for red symbols by adding a constant $\delta$ to the logits of the green symbols, then apply softmax to obtain the final probabilities.

### 4.1.2 UNBIASED REWEIGHTING METHODS

In this section, we present two reweighting methods that satisfy the unbiased property.

$\delta$-**reweight:** Let the watermark code space $\mathcal{E}$ be the interval $[0, 1]$, and let $P_E$ be the uniform probability on $\mathcal{E}$. Leveraging *Inverse Transform Sampling*[1] (Devroye, 1986), we can sample from distribution $P \in \Delta_\Sigma$ using a uniformly distributed random number in $[0, 1]$. Therefore, we have a mapping *sampling*$_P : \mathcal{E} \to \Sigma$. The $\delta$-reweight just returns a delta distribution $R_E(P) = \delta_{sampling_P(E)}$.

It is important to note that while the reweighted distribution for each individual random event $E$ is a delta distribution, the mean output token probabilities remain the original distribution $P$ when considering the randomness of $E$.

$\gamma$-**reweight:** Let the watermark code space $\mathcal{E}$ be the set of all bijective function between vocabularies set $\Sigma$ and a set of indices $[|\Sigma|] = \{1, \ldots, |\Sigma|\}$, where $|\Sigma|$ is the size of vocabularies set $\Sigma$. Essentially, any watermark code $E$ is an indexing function for vocabularies set $\Sigma$, and is also equivalent to a total order on $\Sigma$. Let $P_E$ be the uniform probability on $\mathcal{E}$, it is easy to sample a watermark code $E$ by randomly shuffling the symbol list.

Assume the original distribution is $P_T(t) \in \Delta_\Sigma, \forall t \in \Sigma$. Given the watermark code $E : \Sigma \to [|\Sigma|]$, we construct auxiliary functions $F_I(i) = \sum_{t \in \Sigma} \mathbf{1}(E(t) \le i) P_T(t)$, $F_S(s) = \max(2s - 1, 0)$, $F_{I'}(i) = F_S(F_I(i))$. The $\gamma$-reweight yields new distribution $P_{T'}(t) = F_{I'}(E(t)) - F_{I'}(E(t) - 1)$.

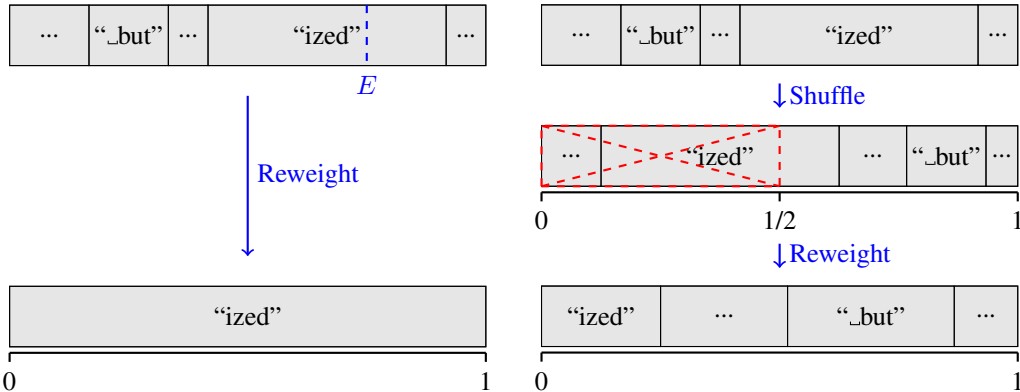

Figure 1: Illustration of $\delta$-reweight.  Figure 2: Illustration of $\gamma$-reweight.

We provide illustrations of the $\delta$-reweight and $\gamma$-reweight methods in Figures 1 and 2. Each block represents a token, and the width represents the probability of that token, so the total length is 1 The left panel shows the $\delta$-reweight method, where each individual random watermark code $E \in [0, 1]$ uniformly sampled from interval $[0, 1]$ corresponds to a specific token according to the horizontal axis, and the reweighted distribution is just a $\delta$ distribution on that token, such that the selected token has 1 probability, and all other vocabulary tokens have a probability of 0. The right panel demonstrates the $\gamma$-reweight method. First, the symbol set is shuffled. Then, the left half of the regions are rejected, and the remaining regions are amplified with a factor of 2.

Both methods are unbiased[1] when considering the randomness of the watermark code $E$. For $\delta$-reweight, we can see that by noticing that the probability of returning a $\delta$ distribution on a token is

---

[1]Detailed definition and rigorous proof can be found in Appendix B

just the original probability on that token, therefore the weighted average of all delta distributions is still the original probability. In the case of $\gamma$-reweight, although certain regions are rejected and the other regions are amplified, every token has the same probability to be in the rejected or amplified region, thus ensuring the unbiased property.

## 4.2 REWEIGHTING FOR AUTOREGRESSIVE MODEL

The reweighting methods presented in the previous section can be applied to single token-generation directly. Given a prefix $\boldsymbol{x}_{1:n}$, the probability distribution for generating a new token without a watermark is denoted as $P_M(\cdot|\boldsymbol{x}_{1:n}) \in \Delta_\Sigma$. For a random watermark code $E$, we sample from a new distribution $P_{M,w}(\cdot|\boldsymbol{x}_{1:n}) = R_E(P_M(\cdot|\boldsymbol{x}_{1:n})) \in \Delta_\Sigma$. If the reweighting function is unbiased, we have $\mathbb{E}_E[R_E(P_M(\cdot|\boldsymbol{x}_{1:n}))] = P_M(\cdot|\boldsymbol{x}_{1:n})$. This ensures that, for an individual unaware of the watermark code, it is impossible to determine whether a new token is sampled directly from $P_M(\cdot|\boldsymbol{x}_{1:n})$ or from $P_{M,w}(\cdot|\boldsymbol{x}_{1:n}; E)$ for a random watermark $E$. However, if the watermark code is known, one can perform statistical hypothesis testing to determine the likelihood of a token being sampled from either distribution.

The main challenge now is constructing the watermark code $E$. Since the LLM generation task is autoregressive, multiple reweighting steps are required, with each step needing a watermark code $E_i$ for reweighting the distribution of token $x_i$.

### 4.2.1 INDEPENDENCE OF WATERMARK CODES

It is crucial that $E_i$ values are independent to ensure the unbiased nature of the entire sequence, rather than just the single-token generation process.

**Theorem 5.** *Given an unbiased reweighting function $(\mathcal{E}, P_E, R)$, if $E_i$ values are i.i.d. with the distribution $P_E$, we have:* $\mathbb{E}_{E_1,\ldots,E_n}[P_{M,w}(\boldsymbol{x}_{1:n}|\boldsymbol{a}_{1:m})] = P_M(\boldsymbol{x}_{1:n}|\boldsymbol{a}_{1:m})$.

If the $E_i$ values are not independent, we cannot guarantee that the generation probability of the entire sequence remains unbiased. As an extreme example, consider a case where all $E_i$ values are identical. Referring to the random bit example in the previous section, assume that the correct distribution is a sequence where each token is a random 0 or 1 with equal probability. Identical $E_i$ values would result in identical token outputs, ultimately producing sequences consisting solely of 0's or 1's, which is clearly biased.

### 4.2.2 CONTEXT CODE

To construct a large number of independent watermark codes $E_i$ during watermarking and to know the used $E_i$ values during watermark detection, we follow an approach similar to Kirchenbauer et al. (2023) by combining the information from the prefix and a secret key to construct $E_i$.

For a single token generation process, given a prefix $x_1, x_2, ..., x_n$, we consider an abstract context code space $C$ and an abstract context code generation function $cc : \Sigma^* \to C$. Based on the prefix, we construct the context code $c_{n+1} = cc(x_1, x_2, ..., x_n)$. Specific examples include using the entire prefix $c_{n+1} = (x_1, x_2, ..., x_n)$, and using the $m$ most recent prefixes $c_{n+1} = (x_{n-m+1}, ..., x_n)$. Our comprehensive framework accommodates diverse context code generation approaches, particularly those that integrate error-correcting mechanisms to augment watermark resilience in the face of text manipulation attacks. Nevertheless, we refrain from delving into these strategies within the confines of this paper and consider it a subject for subsequent investigation.

The final watermark code is defined as $E_i = \hat{E}(c_i, k)$, using a watermark code generation function $\hat{E} : C \times K \to \mathcal{E}$.

**Definition 6.** *Given an unbiased reweighting function $(\mathcal{E}, P_E, R)$ and a context code space $C$, an **unbiased watermark code generation function** is a tuple $(\mathcal{E}, P_E, R, C, K, P_K, \hat{E})$ that satisfies:*

1. *Unbiasedness: $\mathbb{E}_{k\sim P_K}[R_{\hat{E}(c,k)}(P)] = P, \forall P \in \Delta_\Sigma, \forall c \in C$.*

2. *Independence: For any $n$ distinct $c_1, \ldots, c_n \in C$, the values $R_{\hat{E}(c_i,k)}(P)$ are mutually independent.*

**Theorem 7.** *For any unbiased reweighting function and context code space, an unbiased watermark code generation function always exists.*

In practice, pseudorandom numbers can be used to implement the unbiased watermark code generation function in the above theorem. Specifically, the hash value $\mathrm{hash}(c, k)$ can be used as a random

seed to sample $E$ from $P_E$ as an implementation of $E = \hat{E}(c, k)$. In this paper, we employ SHA-256 for hash function and a 1024-bit random bitstring as the key $k$.

An unbiased watermark code generation function ensures that watermark codes $E_i$ are independent with each other if only their context codes are different. During the generation of a sequence, context codes may be repeated, although this is a rare event in practice. If $c_i$ and $c_j$ are equal, then $E_i$ and $E_j$ are also equal, violating the independence of $E_i$. A simple workaround is to skip reweighting for a token when encountering a previously used context code. In other words, we set $P_{M,w}(\cdot | \boldsymbol{a}_{1:m}, \boldsymbol{x}_{1:i-1}) = P_M(\cdot | \boldsymbol{a}_{1:m}, \boldsymbol{x}_{1:i-1})$ if the context code has appeared before.

### 4.3 Framework

---
**Algorithm 1** Watermarking framework

---
1: **Input:** key for watermark $k \in K$, prompt $\boldsymbol{a}_{1:m} \in \Sigma^*$, generate length $n \in \mathbb{N}$, initial code history $cch \in 2^C$, context code function $cc : \Sigma^* \to C$, watermark code generation function $\hat{E} : C \times K \to \mathcal{E}$, and reweighting function $R : \mathcal{E} \times \Delta_\Sigma \to \Delta_\Sigma$.
2: **for** $t = 1, \ldots, n$ **do**
3:     $P_i \leftarrow P_M(\cdot \mid \boldsymbol{a}_{1:m}, \boldsymbol{x}_{1:i-1})$                      ▷ original distribution
4:     $c_i \leftarrow cc(\cdot \mid \boldsymbol{a}_{1:m}, \boldsymbol{x}_{1:i-1})$                       ▷ context code
5:     **if** $c_i \in cch$ **then**
6:         $Q_i \leftarrow P_i$                                ▷ skip the reweighting
7:     **else**
8:         $cch \leftarrow cch \cup \{c_i\}$                     ▷ record history
9:         $E_i \leftarrow \hat{E}(c_i, k)$                        ▷ watermark code
10:        $Q_i \leftarrow R_{E_i}(P_i)$                   ▷ reweighted distribution
11:     Sample the next token $x_i$ using distribution $Q_i$
12: **return** $\boldsymbol{x}_{1:n}$

---

Integrating the tools discussed earlier, we present a general framework for watermarking here. The algorithm for this framework is outlined in Algorithm 1.

We note that our abstract framework requires the specification of two key components in order to be practically implemented: the unbiased reweight function $R_E$ and the context code function $cc$.

## 5 Statistical hypothesis testing for watermark detection

In the previous section, we discussed the process of adding a watermark to a text based on a secret key $k$ and a given prompt $\boldsymbol{a}_{1:m}$. The watermark-embedded text can be sampled from the distribution $P_{M,w}(\boldsymbol{x}_{1:n}|\boldsymbol{a}_{1:m}; k)$. In this section, we focus on the watermark detection task, which is the inverse problem of watermark embedding.

Given a text $\boldsymbol{x}_{1:n}$, the goal of watermark detection is to infer whether it is more likely to be generated from the unmarked distribution $P_M(\boldsymbol{x}_{1:n}|\boldsymbol{a}_{1:m})$ or the marked distribution $P_{M,w}(\boldsymbol{x}_{1:n}|\boldsymbol{a}_{1:m}; k)$. This problem can be formulated as a statistical hypothesis test between two competing hypotheses: $H_0$, which posits that $\boldsymbol{x}_{1:n}$ follows the unmarked distribution, and $H_1$, which posits that $\boldsymbol{x}_{1:n}$ follows the marked distribution.

### 5.1 Score-based tesing

We focus on a particular kind of score-based testing, which assigns a score to each token in the text. The score can be interpreted as the confidence that the token was generated by the watermark model rather than the original model. Scores $s_i$ can be computed based on $\boldsymbol{x}_{1:i}$, in accordance with the autoregressive manner of the generation process.

The total score $S$ is given by $S = \sum_{i=1}^{n} s_i$. A threshold $\hat{S}$ is set such that if $S < \hat{S}$, the null hypothesis $H_0$ is accepted, indicating insufficient evidence to conclude that the text contains a watermark. Otherwise, the null hypothesis is rejected. There are two types of error probabilities associated with this decision process: type I error, which is the probability of incorrectly rejecting the null hypothesis under $H_0$, denoted as $P_{H_0}(S \geq \hat{S})$, and type II error, which is the probability of incorrectly accepting the null hypothesis under $H_1$, denoted as $P_{H_1}(S < \hat{S})$.

To derive theoretical results, we require the scores to have a specific property: under the null hypothesis $H_0$, the exponential momentum of $s_i$ is bounded, conditioned on the preceding context $\boldsymbol{x}_{1,i-1}$. This requirement leads to an upper bound on $\alpha$, the type I error probability.

To derive theoretical results, we require that the scores have a particular property: the exponential moment of $s_i$ under $H_0$ should be bounded, conditioned on the previous text $\boldsymbol{x}_{1,i-1}$. This requirement leads to an upper bound on the type I error rate.

**Theorem 8.** *Given a probability space $(\Omega, \mathcal{A}, P)$ and a $\Sigma$-valued stochastic process $x_i : 1 \leq i \leq n$, as well as an $\mathbb{R}$-valued stochastic process $s_i : 1 \leq i \leq n$, let $\mathcal{F}_i^x := \sigma(x_j \mid 1 \leq j \leq i)$ and $\mathcal{F}_i^s := \sigma(s_j \mid 1 \leq j \leq i)$ be the corresponding filtrations, where $\sigma(\cdot)$ denotes the $\sigma$-algebra generated by random variables. If $\mathcal{F}_i^s \subseteq \mathcal{F}_i^x$ and $\mathbb{E}[\exp(s_i)|\mathcal{F}_{i-1}^x] \leq 1$, then $P(\sum_{i=1}^n s_i \geq t) \leq e^{-t}$.*

Therefore, to ensure that the type I error probability has an upper bound $\alpha$, we can set the threshold $\hat{S}$ as $\hat{S} = -\log(\alpha)$. In the following, we discuss two special scores.

## 5.2 Log Likelihood Ratio (LLR) Score

According to the Neyman-Pearson lemma, the likelihood ratio test is the most powerful test among all tests with the same type I error rate. Specifically, the log-likelihood ratio (LLR) score is defined as $s_i = \log \frac{P_{M,w}(x_i|\boldsymbol{a}_{1:m}, \boldsymbol{x}_{1:i-1};k)}{P_M(x_i|\boldsymbol{a}_{1:m}, \boldsymbol{x}_{1:i-1})}$, and the total score becomes $S = \log \frac{P_{M,w}(\boldsymbol{x}_{1:n}|\boldsymbol{a}_{1:m};k)}{P_M(\boldsymbol{x}_{1:n}|\boldsymbol{a}_{1:m})}$.

We now provide an optimization derivation of the above $s_i$ to gain intuition and set the foundation for the maximin variant of the LLR score in the next section. Let $P_i = P_M(\cdot|\boldsymbol{a}_{1:m}, \boldsymbol{x}_{1:i-1})$, $Q_i = P_{M,w}(\cdot|\boldsymbol{a}_{1:m}, \boldsymbol{x}_{1:i-1};k)$, and let $s_i = S_i(x_i) \in \mathbb{R}$ denote the score corresponding to different $x_i$. Note that $P_i$, $Q_i$, and $S_i$ are all functions with signature $\Sigma \to \mathbb{R}$, therefore equivalent to vectors of dimension $|\Sigma|$. We can define the inner product as $\langle P_i, S_i \rangle = \sum_{x \in \Sigma} P_i(x)S_i(x)$.

The requirement $\mathbb{E}[\exp(s_i)|\mathcal{F}_{i-1}^x] \leq 1$ can be reformulated as $\langle P_i, \exp(S_i) \rangle \leq 1$, where the exponential function is applied element-wise. Instead of minimizing the type II error directly, we aim to maximize the average score under $H_1$, i.e., $\langle Q_i, S_i \rangle$.

The optimization problem becomes $\max_{S_i} \langle Q_i, S_i \rangle$, s.t. $\langle P_i, \exp(S_i) \rangle \leq 1$. The optimal solution is given by $S_i(x) = \log \frac{Q_i(x)}{P_i(x)}$, which recovers the optimal log likelihood ratio score.

## 5.3 Maximin Variant of LLR Score

One major limitation of the LLR score described in the previous section is that when $Q_i(x) = 0$, $S_i(x) = -\infty$. This means that as long as a single token does not come from the watermark model $P_{M,w}$, the score becomes negative infinity, making it impossible to reject the null hypothesis $H_0$.

A more general reason for this issue is that the watermark model $P_{M,w}$ used in the detection process may not exactly match the true distribution of the watermarked text. In practice, potential sources of discrepancy include editing (e.g., a text sampled from $P_{M,w}$ may undergo some degree of editing before being watermark detection) and imperfect estimation of the generation process (e.g., due to lack of knowledge of the exact prompt and temperature used during generation).

To address this problem, we consider a perturbed generation distribution. Instead of the original hypothesis $H_1$, where $\boldsymbol{x}_{1:n}$ follows the watermark distribution $P_{M,w}$, we now assume that $\boldsymbol{x}_{1:n}$ follows a distribution $P'_{M,w}$, which is similar to but not identical to $P_{M,w}$. Specifically, during the generation of each token, the total variation (TV) distance between $Q'_i$ and $Q_i$ is bounded by $d$.

The corresponding new optimization problem is

$$\max_{S_i} \min_{Q'_i \in \Delta_\Sigma, TV(Q'_i, Q_i) \leq d} \langle Q'_i, S_i \rangle, \quad s.t. \langle P_i, \exp(S_i) \rangle \leq 1.$$

Intuitively, the optimal solution for $Q'_i$ in the inner optimization decreases $Q'_i(x)$ when $S_i(x)$ is large and increases $Q'_i(x)$ when $S_i(x)$ is small.

The computation of the maximin solution can be done efficiently in $\widetilde{O}(|\Sigma|)$ time and the specific algorithm is shown in Appendix B.5.

It is important to note that the maximin variant of the LLR score is more robust than the standard LLR score, as it yields higher scores when the text has undergone some degree of editing. However, it is not specifically designed to defend against any attacks.

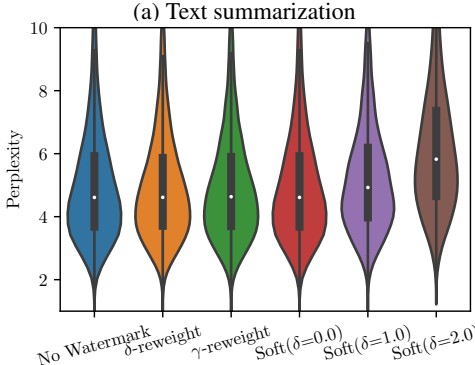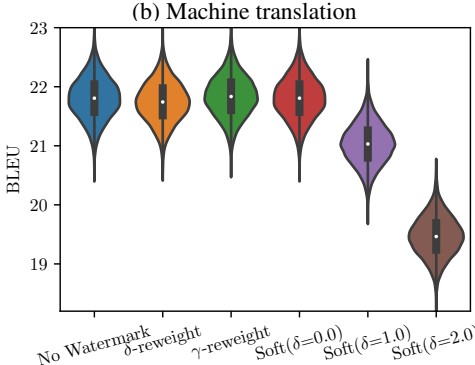

Figure 3: Distribution of perplexity of output for TS and BLEU score for MT.

A hyperparameter $d \in [0, 1]$ that represent the perturbation strength is introduced in the score. Intuitively, if the text to be detected has undergone more editing and deviates further from the distribution $P_{M,w}$, $d$ should be larger. In practice, we recommend using grid search to select the best value of $d$. Assuming there are $A$ candidate values for $d$, corresponding to $A$ different scores $s_i^{(a)}$ ($1 \leq a \leq A$), we can modify Theorem 8 as follows.

**Theorem 9.** *Under the same conditions as Theorem 8, but with multiple scores $s_i^{(a)}$, we have*

$$P\left(\max_{1 \leq a \leq A} \left(\sum_{i=1}^{n} s_i^{(a)}\right) \geq t\right) \leq Ae^{-t}.$$

Thus, when using grid search, the final threshold should be adjusted as $\hat{S} = -\log(\alpha) + \log(A)$. This ensures that the upper bound of the type I error is still $\alpha$.

## 6 EXPERIMENTS

We evaluate the performance of our Unbiased Watermarks on two important applications of seq2seq models: text summarization (TS) and machine translation (MT). For the TS task, we use the BART-large model (Liu et al., 2020) and the CNN-DM (Hermann et al., 2015) corpus as our testing dataset. The MT task involves translating English to Romanian, for which we employ the Multilingual BART (MBart) (Liu et al., 2020) model on the WMT'14 En-Ro corpus. For further details on the experiment setup, please refer to Appendix E.

Table 1: Performance of different watermarking methods on TS and MT. We use F1 scores of BERTScore and scale BERTScore and ROUGE-1 with a factor of 100.

|  | Text summarization | | | Machine translation | |
| --- | --- | --- | --- | --- | --- |
|  | BERTScore ↑ | ROUGE-1 ↑ | Perplexity ↓ | BERTScore ↑ | BLEU ↑ |
| No Watermark | $32.70 \pm 0.08$ | $38.56 \pm 0.09$ | $5.024 \pm 0.018$ | $55.9 \pm 0.3$ | $21.8 \pm 0.3$ |
| $\delta$-reweight | $32.71 \pm 0.08$ | $38.57 \pm 0.09$ | $5.022 \pm 0.018$ | $56.3 \pm 0.3$ | $21.7 \pm 0.3$ |
| $\gamma$-reweight | $32.69 \pm 0.08$ | $38.60 \pm 0.09$ | $5.019 \pm 0.018$ | $56.2 \pm 0.3$ | $21.8 \pm 0.3$ |
| Soft($\delta$=0.0) | $32.70 \pm 0.08$ | $38.56 \pm 0.09$ | $5.024 \pm 0.018$ | $55.9 \pm 0.3$ | $21.8 \pm 0.3$ |
| Soft($\delta$=1.0) | $32.35 \pm 0.08$ | $38.20 \pm 0.09$ | $5.313 \pm 0.018$ | $55.1 \pm 0.3$ | $21.0 \pm 0.3$ |
| Soft($\delta$=2.0) | $31.21 \pm 0.08$ | $37.17 \pm 0.08$ | $6.253 \pm 0.022$ | $53.8 \pm 0.3$ | $19.5 \pm 0.3$ |

Our primary focus is to compare the performance of our proposed unbiased watermarking methods including the $\delta$-reweight and $\gamma$-reweight, with the soft-red-list method presented by Kirchenbauer et al. (2023). The watermark strength in the soft-red-list approach is controlled by a parameter $\delta$.

The quality of output post-watermarking is presented in Table 1. We observed that the output quality remains unaffected by our unbiased watermark methods, both for the $\delta$-reweight and $\gamma$-reweight, irrespective of the task and metric. Conversely, the soft-red-list method, when $\delta = 0$, does not introduce any watermark and hence does not affect output quality. However, for $\delta > 0$, it significantly impairs the quality of output.

Table 3: Text sampled from OPT-6.7B, with and without watermarks. For "No watermark" (NW), the score is computed based on $\delta$-reweight. When watermarks are included, the corresponding reweighting function is used for computing score. The optimal perturbation strengths $d$ obtained by grid search are $0.9, 0.0, 0.0$ for three outputs respectively.

| Prompt | What is a watermark? What's the purpose of it? | score | p-value[2] |
|---|---|---|---|
| NW | Why don't you want it on there? I'm confused..\nI think he meant to say logo, since he wrote "watermark", so the first word must be a typo.\nYes! Exactly typo. Sorry, English is not my native language.. Thanks for the explanation! | 0.30 | 8.14 |
| $\delta$-RW | \nIt is supposed to be watermarking the pictures that you took with your phone i think. So, so you can share your pictures and not take credit for them. | 75.9 | 1.2e−32 |
| $\gamma$-RW | \nA watermark is a small image or logo (often in square pixels) that is placed over the larger, original image. It serves primarily to distinguish copyright or ownership of large images (such as banners and logos) and, on rare occasion, to identify small images (such as thumbnail images for blog posts and pictures). | 32.9 | 5.7e−14 |

Figure 3 provides a more intuitive depiction of the score distributions. It is evident that our unbiased watermark methods not only ensure that the mean performance remains unaffected but also that the performance distribution is stable. Conversely, the soft-red-list method shows a notable performance decrease.

In terms of watermark detection, we compute score associated with each token. The mean and variance of score per token for TS and MT are presented in Table 2. As a heuristic, if the sum of the scores for all tokens in a sentence reaches 10, a p-value of less than 0.0005 is ensured. If the sum score hits 20, the p-value must be less than 3e−8.

Table 2: Mean and variance of score per token for different reweighting methods and different tasks.

| | Text summarization | Machine translation |
|---|---|---|
| $\delta$-RW | $0.8784 \pm 1.4354$ | $0.4192 \pm 1.1361$ |
| $\gamma$-RW | $0.2207 \pm 0.3678$ | $0.1056 \pm 0.2916$ |

Additionally, we provide an example of watermarking applied to a completion task in Table 3. It visually demonstrates the score distribution across tokens: positive scores are represented in green, and negative ones in red. The intensity of the color corresponds to the magnitude of the score, with darker shades representing larger absolute values.

## 7    RELATED WORK

The idea of watermarking text has been widely explored by many researchers (Cox et al., 2007; Kamaruddin et al., 2018; Podilchuk & Delp, 2001; Potdar et al., 2005; Atallah et al., 2001; Jalil & Mirza, 2009; Stefan et al., 2000; Petitcolas et al., 1999), even before the advent of large language models.

Recent advancements in generative models have opened new possibilities for directly generating watermarked results. Two relevant prior works in this domain are by Kirchenbauer et al. (2023) and Aaronson (2022). Various concurrent studies (Christ et al., 2023; Kuditipudi et al., 2023; Wang et al., 2023b; Yoo et al., 2023b) have further enriched this domain. Due to space constraints, we moved the in-depth analysis and other related work to Section A.

## 8    CONCLUSION

Overall, this paper provides a novel framework of watermarking for language models, demonstrating that it is possible to use watermark to protect intellectual property and monitor potential misuse without compromising the quality of the generated text. This research serves as a valuable foundation for future work in the field of watermarking for large language models.

---

[2]This is an upper bound computed based on Theorem 9. The upper bound could be larger than 1, but this does not necessarily imply that the p-value exceeds 1.

# 9 ETHICS STATEMENTS

Our unbiased watermark has removed major hurdles for large-scale application of watermarks. The two primary obstacles previously were the potential for watermarks to degrade the quality of output and the possibility for users to discern the presence of watermarks. Our method addresses both of these issues thoroughly.

## 9.1 IMPACT ANALYSIS

**Traceability and accountability**  Traceability refers to the ability to trace back the origin of a text. Any watermarking method, including method in this paper, contribute to traceability. In an era of misinformation and disinformation, this allows for holding providers accountable for the content generated by their models.

**Identifying model-generated texts**  Watermarking method can be used to distinguish which texts are generated by the models. This prevents unnecessary training on the data generated by the models themselves.

**Ownership**  Watermarking method can help provide evidence in situations where a provider claims ownership over a generated text (Sun et al., 2023).

**Data privacy concerns**  The use of different watermarks, if applied discretionarily, could potentially link generated text back to a specific user or request. This could be seen as a breach of users' privacy, raising important data privacy concerns.

**Manipulation and removal of watermarks**  The ongoing development of techniques to manipulate or remove watermarks could lead to an "arms race" between providers attempting to secure their watermarks and users trying to remove them.

## 9.2 ETHICAL CONSIDERATIONS

There are several ethical considerations in the pursuit of watermarking technology.

**Consent**  Users have the right to be informed about the use of watermarks and should have the option to opt-out.

**Transparency**  Providers should be transparent about the use of watermarks, including information on what is embedded within these watermarks and how it's used. If the watermarks contain identifying information, providers should clearly state who can access this information and take appropriate measures to prevent potential misuse.

**Fair use**  The application of our watermarking technique should not interfere with the legitimate use of the service by users.

Our watermarking method does not degrade the quality of the output, ensuring the values of fair use are upheld. However, it also introduces a potentially challenging issue.

Due to the undetectable nature of our technique, every user might have to assume that the service they are using has been watermarked, as it cannot be disproved. This raises challenging questions on how to ensure consent and transparency.

## 9.3 CONCLUSION

Our unbiased watermarking method brings improved traceability and attribution and ensures that fair use is not compromised. However, it also poses significant challenges in data privacy, transparency, and consent. Any implementation of this system needs to be done thoughtfully and ethically, with clear communication to users about how it works and what it means for them.

ACKNOWLEDGMENTS

This work was partially supported by NSF IIS 2347592, 2347604, 2348159, 2348169, DBI 2405416, CCF 2348306, CNS 2347617.

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

## A  RELATED WORKS

### A.1  TEXT WATERMARKING

The idea of watermarking text has been widely explored by many researchers (Cox et al., 2007; Kamaruddin et al., 2018; Podilchuk & Delp, 2001; Potdar et al., 2005; Atallah et al., 2001; Jalil & Mirza, 2009; Stefan et al., 2000; Petitcolas et al., 1999), even before the advent of large language models. Several techniques involve editing existing text to add a watermark, such as changing synonyms (Topkara et al., 2005; 2006c; Chiang et al., 2004; Venugopal et al., 2011; Yang et al., 2022) or visually indistinguishable words (Rizzo et al., 2019), altering sentence structures (Topkara et al., 2006b;a; Meral et al., 2009), and employing neural networks (He et al., 2022a;b; Yoo et al., 2023a).

Recent advancements in generative models have opened new possibilities for directly generating watermarked results. Two prior works in this domain are by Kirchenbauer et al. (2023) and Aaronson (2022). Kirchenbauer et al.'s pioneering work, which uses the previous context to generate watermarked tokens, heavily influences our approach. However, their watermarking technique can introduce bias to the output, leading to performance degradation. Our work addresses this limitation by applying unbiased reweighting and recording context code history.

Aaronson (2022) have talked about using a pseudo-random cryptographic function for watermarking, but the details are not disclosed, making it challenging to conduct a direct comparison. Aaronson's "cryptographic pseudorandom function" could be a special case of reweighting function in this paper. However, in his blog, there is no apparent structure akin to "context code history", a mechanism that plays a crucial role in our work to ensure n-shot-undetectability. Therefore, it remains uncertain whether Aaronson's technique could offer a similar theoretical guarantee of n-shot-undetectability as ours. Additionally, it is not clear if their method provides an upper bound on type I error, like Theorem 8.

Several concurrent studies have explored methods to reduce the bias in watermarking. (Christ et al., 2023) depends on computational power to ensure that an attacker cannot efficiently detect watermarks. This approach presents a different trade-off from our work; while we rely on additional watermark storage, we can strictly guarantee n-shot undetectability, regardless of the computational resources available to the attacker. Later, Kuditipudi et al. (2023) builds a watermark based on a watermark key sequence. However, when the generated content length exceeds the length of the watermark key sequence, it may use the same key sequence, resulting in a compromise of strict unbiasedness.

Additionally, there has been research focus on multi-bit watermarking such as Wang et al. (2023b) and Yoo et al. (2023b).

## A.2 ATTACKS ON WATERMARKS

Alongside the development of watermarking technologies, various methods to modify and remove these watermarks and their countermeasures have also been explored. These include attacks based on invisible characters and homoglyphs (Gabrilovich & Gontmakher, 2002; Helfrich & Neff, 2012; Pajola & Conti, 2021; Boucher et al., 2022), generative attacks such as those that prompted the model to change its output in a predictable and easily reversible way (Kirchenbauer et al., 2023), and specific instances such as the emoji attack (Goodside, 2023), and paraphrasing attacks (Sadasivan et al., 2023; Krishna et al., 2023).

## A.3 STEGANOGRAPHY IN TEXT

Steganography hides information in text primarily for secret communication. It bears similarities to watermarking in that it seeks to conceal information. However, while watermarking only needs to detect the presence of a watermark, steganography must recover all embedded information. Many approaches have tried to edit existing text, through rule-based transformations (Wilson et al., 2014; 2015; Wilson & Ker, 2016), synonym-based methods (Shirali-Shahreza & Shirali-Shahreza, 2008), and more recently, neural network-based methods (Abdelnabi & Fritz, 2021; Ueoka et al., 2021). Information can also be embedded directly during generation (Fang et al., 2017; Dai & Cai, 2019; Ziegler et al., 2019).

## A.4 WATERMARKING MODELS

Watermarking has also been applied to models themselves to protect intellectual property rights and to guard against model stealing or extraction (Jia et al., 2021; Boenisch, 2021; Zhao et al., 2023). The aim here is to gather evidence through inference services (Li et al., 2019; Zhang et al., 2018) and can be accomplished by adding backdoors to models (Adi et al., 2018; Gu et al., 2017; 2022). While they are similar to text watermarking in that they embed information without impacting fair use, the focus is on tracing the model rather than the text.

## A.5 DETECTING MACHINE-GENERATED TEXT

The objective of detecting machine-generated text lies in discerning whether a given text has been produced by an algorithm or written by a human. Such detection is crucial to prevent misuse and a substantial body of research has explored this area (Zellers et al., 2019; Ippolito et al., 2019; Crothers et al., 2022; Jawahar et al., 2020; Tan et al., 2020; Tay et al., 2020; Tang et al., 2023; Wang et al., 2023a). However, the task has become increasingly challenging due to the continual improvement in language models and the advent of adversarial attacks (Gambini et al., 2022; Wolff & Wolff, 2020; Sadasivan et al., 2023). The difference between this and text watermarking is that watermarking is employed to differentiate whether a text is generated by a particular model or provider, yet the detection of machine-generated text is not concerned with a specific model.

## B    DETAILED DEFINITION AND ADDITIONAL PROOFS

### B.1    DETAILED DEFINITION AND ADDITIONAL PROOFS FOR SECTION 4.1

**Definition 10** (hard/soft-red-list reweighting (Kirchenbauer et al., 2023)). *Given two hyperparameters $0 \leq \gamma \leq 1$ and $\delta \geq 0$, let the watermark code space be $\mathcal{E} = \{E \in \{0,1\}^{\Sigma} \mid |E^{-1}(1)| = \lfloor \gamma |\Sigma| \rfloor \}$, such that $f$ maps $\gamma$-portion of the tokens in $\Sigma$ to 1 (which interpreted as "green") and the other portion to 0 (which interpreted as "red"), and let $P_E$ to be the uniform distribution on space $\mathcal{E}$. For any watermark code $E$, and for any token distribution $P \in \Delta_{\Sigma}$, the output distribution of the hard-red-list reweighting function on a token $t \in \Sigma$ is defined by $R_E(P)(t) = \frac{E(t)P(t)}{\sum_{t \in \Sigma} E(t)P(t)}$ assuming $\sum_{t \in \Sigma} E(t)P(t) > 0$. The soft-red-list reweighting function is defined by $R_E(P)(t) = \frac{\exp\{\log P(t) + \delta E(t)\}}{\sum_{t \in \Sigma} \exp\{\log P(t) + \delta E(t)\}}$, where $\delta > 0$ is a fixed constant.*

**Theorem 11.** *Hard-red-list and soft-red-list reweighting functions are biased.*

*Proof.* We first show the hard-red-list reweighting is biased. For $\gamma = 0.5$, consider $\Sigma = \{a, b\}$, $P(a) = 0.9, P(b) = 0.1$, we have

$$R_E(P)(a) = \frac{1}{2} \times \frac{P(a)}{P(a)} + 0 \times \frac{0}{P(b)} = 0.5 \neq 0.9 = P(a).$$

We then show the soft-red-list reweighting is biased. For $\gamma = 0.5$, consider $\Sigma = \{a, b\}$, $P(a) = 0.9, P(b) = 0.1$, we have

$$R_E(P)(a) = \frac{1}{2} \times \frac{e^{\delta} P(a)}{e^{\delta} P(a) + P(b)} + \frac{1}{2} \times \frac{P(a)}{P(a) + e^{\delta} P(b)}.$$

It is easy to verify that for any $\delta > 0$, we have $R_E(P)(a) < P(a)$.

Thus, hard/soft-red-list reweighting are both biased.    $\square$

**Definition 12** ($\delta$-reweight). *Let the watermark code space $\mathcal{E}$ be the interval $[0, 1]$, and let $E$ be uniformly distributed on $\mathcal{E}$. Given an arbitrary token distribution $P \in \Delta_{\Sigma}$, let $B$ be a bijection between $\Sigma$ and $[|\Sigma|]$, we construct a cumulative density function of $P$ w.r.t. $B$ by $F_P(t; B) = \sum_{t' \in \Sigma} \mathbf{1}(B(t') \leq B(t)) P(t'), \forall t \in \Sigma$. Then we can define a mapping $sampling_P : \mathcal{E} \rightarrow \Sigma$,*

$$sampling_P(E) = B^{-1}(I(E)),$$

*where*

$$I(E) = \min_t B(t) \ s.t. \ E \leq F_P(t; B),$$

*The $\delta$-reweight function is defined by $R_E(P) := \delta_{sampling_P(E)}$.*

**Definition 13** ($\gamma$-reweight). *Let the watermark code space $\mathcal{E}$ be the set of all bijective function between vocabularies set $\Sigma$ and a set of indices $[|\Sigma|] = \{1, \dots, |\Sigma|\}$, where $|\Sigma|$ is the size of vocabularies set $\Sigma$. Assume the original distribution is $P_T(t) \in \Delta_{\Sigma}, \forall t \in \Sigma$. Given the watermark code $E : \Sigma \rightarrow [|\Sigma|]$, we define*

$$A_E(i) := \max \left\{ 2 \left( \sum_{t \in \Sigma} \mathbf{1}(E(t) \leq i) P_T(t) \right) - 1, 0 \right\},$$

*where $\mathbf{1}(E(t) \leq i) = 1$ when $E(t) \leq i$ otherwise $\mathbf{1}(E(t) \leq i) = 0$. We define $P_{T'(E)}(t) := A_E(E(t)) - A_E(E(t) - 1)$. It's easy to verify $P_{T'(E)}$ is a distribution by $\forall t \in \Sigma, P_{T'(E)}(t) \geq 0$ and $\sum_{t \in \Sigma} P_{T'(E)}(t) = 1$. Thus, $\gamma$-reweight function is defined by $R_E(P_T) := P_{T'(E)}$.*

**Theorem 14.** *Both $\delta$-reweight and $\gamma$-reweight are unbiased reweighting functions.*

*Proof.* According to Definition 4, we need to show $\mathbb{E}_E[R_E(P)] = P$ for arbitrary $P \in \Delta_{\Sigma}$.

For $\delta$-reweight, we have $R_E(P) = \delta_{sampling_P(E)}$ and $E$ is uniformly distributed on $[0, 1]$. Thus, we only need to show $\forall t \in \Sigma$, $\mathbb{E}_E[\delta_{sampling_P(E)}(t)] = P(t)$.

$$
\begin{aligned}
\mathbb{E}_E[\delta_{sampling_P(E)}(t)] &= \int_0^1 \mathbf{1}(sampling_P(e) = t)\, de, \\
&= \int_0^1 \mathbf{1}(I(e) = B(t))\, de, \\
&= \begin{cases} F_P(t; B) - F_P(B^{-1}(B(t) - 1); B) & B(t) > 1 \\ F_P(t; B) & B(t) = 1 \end{cases} \\
&= P(t).
\end{aligned}
\tag{1}
$$

For $\gamma$-reweight, we need to show $\forall t \in \Sigma$, $\mathbb{E}_E[R_E(P_T)(t)] = P_T(t)$

$$
\begin{aligned}
\mathbb{E}_E[R_E(P_T)(t)] &= \mathbb{E}_E[P_{T'(E)}(t)] \\
&= \mathbb{E}_E[A_E(E(t)) - A_E(E(t) - 1)].
\end{aligned}
\tag{2}
$$

Denoted by $g_E(i) = 2\left(\sum_{t' \in \Sigma} \mathbf{1}(E(t') \le i)P_T(t')\right) - 1$. $\forall E \in \mathcal{E}$, we consider the reserved order $E^r$ of $E$, we have $E(t) + E^r(t) = n + 1$ and

$$
g_E(E(t)) + g_{E^r}(E^r(t) - 1) = 2\left(\sum_{t' \in \Sigma} [\mathbf{1}(E(t') \le E(t)) + \mathbf{1}(E(t') \ge E(t) + 1)]P_T(t')\right) - 2 = 0.
$$

So we have

$$
\begin{aligned}
&A_E(E(t)) - A_E(E(t) - 1) + A_{E^r}(E^r(t)) - A_{E^r}(E^r(t) - 1) \\
=& \max\{g_E(E(t)), 0\} - \max\{g_E(E(t) - 1), 0\} + \max\{g_E^r(E^r(t)), 0\} - \max\{g_E^r(E^r(t) - 1), 0\} \\
=& g_E(E(t))\mathbf{1}(g_E(E(t)) > 0) - g_{E^r}(E^r(t) - 1)\mathbf{1}(g_{E^r}(E^r(t) - 1) > 0) + \\
& g_{E^r}(E^r(t))\mathbf{1}(g_{E^r}(E^r(t)) > 0) - g_E(E(t) - 1)\mathbf{1}(g_E(E(t) - 1) > 0) \\
=& g_E(E(t))\mathbf{1}(g_E(E(t)) > 0) + g_E(E(t))\mathbf{1}(g_E(E(t)) < 0) - \\
& g_E(E(t) - 1)\mathbf{1}(g_E(E(t) - 1) < 0) - g_E(E(t) - 1)\mathbf{1}(g_E(E(t) - 1) > 0) \\
=& g_E(E(t)) - g_E(E(t) - 1) \\
=& 2P_T(t),
\end{aligned}
\tag{3}
$$

which yields

$$
\begin{aligned}
\mathbb{E}_E[R_E(P_T)](t) &= \mathbb{E}_E[A_E(E(t)) - A_E(E(t) - 1)]. \\
&= \frac{1}{2}\left(\mathbb{E}_E[A_E(E(t)) - A_E(E(t) - 1)] + \mathbb{E}_{E^r}[A_{E^r}(E^r(t)) - A_{E^r}(E^r(t) - 1)]\right). \\
&= \frac{1}{2}\mathbb{E}_E[2P_T(t)] \\
&= P_T(t).
\end{aligned}
\tag{4}
$$

$\square$

## B.2 Additional proofs for Section 4.2

*Proof of Theorem 5.* We have

$$
\begin{aligned}
&\mathbb{E}_{E_1,\ldots,E_n}[P_{M,w}(\boldsymbol{x}_{1:n}|\boldsymbol{a}_{1:m})] \\
=&\mathbb{E}_{E_1,\ldots,E_{n-1}}[\mathbb{E}_{E_n}[P_{M,w}(\boldsymbol{x}_{1:n}|\boldsymbol{a}_{1:m})]] \\
=&\mathbb{E}_{E_1,\ldots,E_{n-1}}[\mathbb{E}_{E_n}[P_{M,w}(x_n|\boldsymbol{a}_{1:m}, \boldsymbol{x}_{1:n-1})]P_{M,w}(\boldsymbol{x}_{1:n-1}|\boldsymbol{a}_{1:m})] \\
=&\mathbb{E}_{E_n}[P_{M,w}(x_n|\boldsymbol{a}_{1:m}, \boldsymbol{x}_{1:n-1})]\mathbb{E}_{E_1,\ldots,E_{n-1}}[P_{M,w}(\boldsymbol{x}_{1:n-1}|\boldsymbol{a}_{1:m})] \\
=&P_M(x_n|\boldsymbol{a}_{1:m}, \boldsymbol{x}_{1:n-1})\mathbb{E}_{E_1,\ldots,E_{n-1}}[P_{M,w}(\boldsymbol{x}_{1:n-1}|\boldsymbol{a}_{1:m})],
\end{aligned}
$$

where the second last step uses the independence of the $E_i$ values and the last step uses the unbiasedness of the reweighting function. Repeating the same argument for the remaining $E_i$ values, we obtain

$$\mathbb{E}_{E_1,\ldots,E_n}[P_{M,w}(\boldsymbol{x}_{1:n}|\boldsymbol{a}_{1:m})] = P_M(\boldsymbol{x}_{1:n}|\boldsymbol{a}_{1:m}).$$

$\square$

*Proof of Theorem 7.* Given a watermark code space $\mathcal{E}$ and a watermark code distribution $P_E(e)$, we construct a key space $K = \mathcal{E}^C$, where each key $k$ is a function from the context code space to the watermark code space. The random key probability density function is defined as $P_K(k) = \prod_{c \in C} P_E(k(c))$.

This construction forms a particular instance of an unbiased watermark code generation function.

$\square$

### B.3 DETAILED THEORY FOR SECTION 4.3

**Corollary 15.** *For every generation request by a user, Algorithm 1 can provide a generation result. This generation service is $n$-shot undetectability for any $n \in \mathbb{N}^+$ if the unbiased watermark code generation function is employed, and the context code history is continuously recorded. Specifically, the context code history cch is updated after each invocation of Algorithm 1, and the resulting context code history is used as the initial context code history for the next invocation.*

*On the other hand, if the context code history is reset after every generation task, the generation service can only guarantee $1$-shot undetectability.*

*Proof.* The key design element in this service is the context code history. By maintaining the context code history throughout the generation process, we can ensure that each time the reweighting is performed, the context code is unique, i.e., it has not appeared in any previous generation tasks. According to the properties of the unbiased watermark code generation function in Definition 6, this guarantees that the watermark codes generated during each reweighting are independent of previously generated watermark codes. As a result, the final distribution is unbiased, and $n$-shot undetectability is achieved.

However, if the context code history is reset after every generation task, it is possible for two invocations of Algorithm 1 to produce the same context code, leading to the same watermark code. Consequently, $n$-shot undetectability cannot be guaranteed for $n > 1$, and the generation service can only provide 1-shot undetectability. $\square$

A straightforward variant of the above approach exists in the form of a batch variant. If the batch size is set to $b$ and the context code history is reset after each batch, the system can ensure $b$-shot undetectability.

### B.4 PROOF OF TAILED BOUNDS IN SECTION 5

*Proof of Theorem 8.*

$$\mathbb{E}\left[\exp\left(\sum_{i=1}^n s_i\right)\right] = \mathbb{E}\left[\exp\left(\sum_{i=1}^{n-1} s_i\right)\mathbb{E}[\exp(s_n)|\mathcal{F}_{n-1}^x]\right]$$

$$\leq \mathbb{E}\left[\exp\left(\sum_{i=1}^{n-1} s_i\right)\right] \leq \cdots \leq 1,$$

where the abbreviation in the last step means applying similar inequalities multiple times.

By applying the Chernoff bound, we obtain the desired result. $\square$

*Proof of Theorem 9.* From Theorem 3, we know that

$$P\left(\sum_{i=1}^n s_i^{(a)} \geq t\right) \leq e^{-t}.$$

Thus,

$$P\left(\max_{1\le a\le A}\left(\sum_{i=1}^{n}s_i^{(a)}\right)\ge t\right)\le\sum_{1\le a\le A}P\left(\sum_{i=1}^{n}s_i^{(a)}\ge t\right)\le Ae^{-t}.$$

$\square$

## B.5 Details on maximin variant of LLR score

### B.5.1 Derivation of the solution

Recall that we are dealing with the maximin problem given as:

$$\max_{S_i}\quad\min_{Q_i'\in\Delta_\Sigma,TV(Q_i',Q_i)\le d}\quad\langle Q_i',S_i\rangle$$
$$s.t.\quad\langle P_i,\exp(S_i)\rangle\le 1.$$

We can find a relaxation by replacing the constraint $Q_i'\in\Delta_\Sigma$ with $\sum_{x\in\Sigma}Q_i'(x)=1$ and no longer requiring $Q_i'(x)\ge 0$. Thus, we obtain the following inequality:

$$\min_{Q_i'\in\Delta_\Sigma,TV(Q_i',Q_i)\le d}\langle Q_i',S_i\rangle\ge\min_{Q_i',\sum_{x\in\Sigma}Q_i'(x)=1,TV(Q_i',Q_i)\le d}\langle Q_i',S_i\rangle.$$

The new maximin problem becomes:

$$\max_{S_i}\quad\min_{Q_i',\sum_{x\in\Sigma}Q_i'(x)=1,TV(Q_i',Q_i)\le d}\quad\langle Q_i',S_i\rangle$$
$$s.t.\quad\langle P_i,\exp(S_i)\rangle\le 1.$$

This relaxation is tight, meaning it does not affect the final maximin optimal solution. This is because, even though the relaxed problem does not require $Q_i'(x)\ge 0$, the maximin problem's optimal solution $S_i^*$ and $Q_i'^*$ must satisfy $Q_i'^*(x)\ge 0$. Otherwise, $S_i^*(x)$ could be further reduced, implying that $S_i^*(x)$ is not an optimal solution and leading to a contradiction.

The inner optimization of the relaxed problem can be solved directly:

$$\min_{Q_i',\sum_{x\in\Sigma}Q_i'(x)=1,TV(Q_i',Q_i)\le d}\langle Q_i',S_i\rangle=\langle Q_i,S_i\rangle+d\left(\min_x S_i(x)-\max_x S_i(x)\right).$$

This leads to the new maximization optimization problem:

$$\max_{S_i}\quad\langle Q_i,S_i\rangle+d\left(\min_x S_i(x)-\max_x S_i(x)\right)$$
$$s.t.\quad\langle P_i,\exp(S_i)\rangle\le 1.$$

We can find the KKT conditions for this optimization problem by rewriting it as follows:

$$\max_{S_i}\quad\langle Q_i,S_i\rangle+d(\max S_i-\min S_i)$$
$$s.t.\quad\langle P_i,\exp(S_i)\rangle\le 1,$$
$$\max S_i\ge S_i(x),$$
$$\min S_i\le S_i(x).$$

Let the Lagrangian be

$$L=\max_{S_i}\langle Q_i,S_i\rangle+d(\min S_i-\max S_i)$$
$$+\lambda(1-\langle P_i,\exp(S_i)\rangle)$$
$$+\langle u,\max S_i-S_i\rangle$$
$$+\langle v,S_i-\min S_i\rangle.$$

Then, the KKT conditions are:

$$\frac{\partial L}{\partial S_i(x)} = [Q_i(x) - u(x) + v(x)] - \lambda P_i(x) \exp(S_i(x)) = 0,$$

$$\frac{\partial L}{\partial \max S_i} = -d + \sum_{x \in \Sigma} u(x) = 0,$$

$$\frac{\partial L}{\partial \min S_i} = d - \sum_{x \in \Sigma} v(x) = 0,$$

$$\lambda(1 - \langle P_i, \exp(S_i) \rangle) = 0,$$
$$\langle u, \max S_i - S_i \rangle = 0,$$
$$\langle v, S_i - \min S_i \rangle = 0.$$

We can solve for the value of $\lambda$:

$$\sum_{x \in \Sigma} \frac{\partial L}{\partial S_i(x)} = [1 - d + d] - \lambda \sum_{x \in \Sigma} P_i(x) \exp(S_i(x)) = 0.$$

Note that $\lambda$ cannot be 0, so the fourth KKT condition implies $\langle P_i, \exp(S_i) \rangle = 1$. Consequently, the above equation implies $\lambda = 1$.

The final solution is given by:

$$S_i(x) = \log \frac{Q_i(x) - u(x) + v(x)}{P_i(x)},$$
$$u(x) \neq 0 \text{ iff } S_i(x) = \max_x S_i(x),$$
$$v(x) \neq 0 \text{ iff } S_i(x) = \min_x S_i(x),$$
$$\sum_{x \in \Sigma} u(x) = \sum_{x \in \Sigma} v(x) = d.$$

### B.5.2 COMPUTING THE SOLUTION

Let

$$X_{\max} = \{x \in \Sigma \mid S_i(x) = \max_x S_i(x)\},$$
$$X_{\min} = \{x \in \Sigma \mid S_i(x) = \min_x S_i(x)\}.$$

If $x \notin X_{\max} \cup X_{\min}$, then we have

$$S_i(x) = \log \frac{Q_i(x)}{P_i(x)}.$$

If $x \in X_{\max}$, then we have

$$\max_x S_i(x) = S_i(x) = \log \frac{Q_i(x) - u(x) + v(x)}{P_i(x)}.$$

Summing over all $x \in X_{\max}$, and noting that $\sum_{x \in X_{\max}} u(x) = d$, we obtain:

$$\max_x S_i(x) = \log \frac{\sum_{x \in X_{\max}} Q_i(x) - d + \sum_{x \in X_{\max}} v(x)}{\sum_{x \in X_{\max}} P_i(x)}.$$

Similarly,

$$\min_x S_i(x) = \log \frac{\sum_{x \in X_{\min}} Q_i(x) - \sum_{x \in X_{\min}} u(x) + d}{\sum_{x \in X_{\min}} P_i(x)}.$$

When $\sum_{x \in X_{\min}} u(x) \neq 0$, it implies that there exists an $x \in X_{\min}$ such that $x \in X_{\max}$, which in turn implies that $\max_x S_i(x) = S_i(x) = \min_x S_i(x)$. In this case, the score is trivial, with $S_i(x) = 0$ for all $x \in \Sigma$.

Thus, the computation of the maximin solution reduces to finding $X_{\max}$ and $X_{\min}$, which can be computed in $\widetilde{O}(|\Sigma|)$ time. A pseudocode is shown in Algorithm 2.

Note that the provided pseudocode is not a real implementation but serves as a schematic representation of the algorithm. In our experimental implementation, we took into consideration the effective precision of computer floating-point numbers. To ensure numerical stability and prevent NaNs, we implemented the algorithm in log space. This makes the algorithm more complex, and additionally, we designed the algorithm with grid search by reusing previous computation results for acceleration. We also implemented such algorithm with tensor operator for efficient computation on GPU. For more details, please refer to the source code.

**Algorithm 2** Computation of maximin variant of LLR score

```python
import numpy as np

def get_max_lr(P: np.ndarray, Q: np.ndarray, d: float) -> float:
    """Get $\max_x \exp(S(x))$"""
    indexes = sorted(range(len(P)), key=lambda i: Q[i] / P[i], reverse=True)

    sum_Q = 0.0
    sum_P = 0.0

    def _lr():
        nonlocal sum_Q, sum_P
        if sum_Q <= d:
            return 0.0
        else:
            return (sum_Q - d) / sum_P

    lr = _lr()

    for i in indexes:
        if Q[i] / P[i] < lr:
            break
        sum_Q += Q[i]
        sum_P += P[i]
        lr = _lr()
    return lr

def get_min_lr(P: np.ndarray, Q: np.ndarray, d: float) -> float:
    """Get $\min_x \exp(S(x))$"""
    indexes = sorted(range(len(P)), key=lambda i: Q[i] / P[i])

    sum_Q = 0.0
    sum_P = 0.0

    def _lr():
        nonlocal sum_Q, sum_P
        return (sum_Q + d) / sum_P

    lr = _lr()

    for i in indexes:
        if Q[i] / P[i] > lr:
            break
        sum_Q += Q[i]
        sum_P += P[i]
        lr = _lr()
```

```
        return lr

def get_S(P: np.ndarray, Q: np.ndarray, d: float) -> np.ndarray:
    max_lr = get_max_lr(P, Q, d)
    min_lr = get_min_lr(P, Q, d)
    lr = Q / P
    if max_lr <= min_lr:
        return np.zeros_like(p)
    return np.log(np.clip(lr, min_lr, max_lr))
```

## C  ADDITIONAL DISCUSSION

**Performance without context code history**   Despite that "context code history" is necessary to ensure $n$-shot-undetectable, it's possible to bypass this requirement, and always execute steps 9 and 10 in Algorithm 1. In many instances, this won't significantly degrade the performance of downstream tasks, as the probability of context code collision is low. However, if one chooses to neglect the context code history, they effectively waive the theoretical guarantee of $n$-shot-undetectability and potentially expose themselves to corner cases that could notably undermine the task performance. Moreover, users could specifically construct test cases that check for the existence of watermarks. For instance, prompts like "Generate a random bit (0 or 1):" or "Generate a random bit sequence, with five dots between every two digits:" would yield incorrect results in the absence of context code history.

**Computation of logits during detection**   The watermark detection methods in Sections 5.2 and 5.3 relies on the output probability distribution $P_M$. Ideally, the $P_M$ used during detection should be the same as the one during generation. However, this may not always be possible. Language model logits depend on various parameters like the prompt, the temperature and sampling policy used during generation, etc., which might not be accessible during watermark detection. For instance, $P_M$ depends on the prompt, but during detection, we might only have the text to be examined and not the prompt from which it was generated.

In such circumstances, we can only resort to using another distribution $P'_M$ as an estimation of $P_M$. For instance, if the prompt is missing during detection, we can set the prompt to an empty string and then calculate the language model probabilities. In a machine translation task, one could translate the output back to the input language and use that as input. In practice, there's likely to be a disparity between $P'_M$ and $P_M$, which could lead to a drop in score. We discuss in detail how the score is affected by changes in logits in Appendix F.2.

**Cost of likelihood computation**   The detection methods in Sections 5.2 and 5.3 require the output probability distribution $P_M$. This comes at a computational cost: it's more computationally expensive than red list-based methods proposed by Kirchenbauer et al. (2023), as it involves a language model. However, the cost is much less than a generation, as it only requires a single forward pass.

On the other hand, our framework also supports likelihood-agnostic detection methods, which have their own pros and cons. We present a detailed comparison of likelihood-based and likelihood-agnostic methods and provide an example in Appendix D.

**Perturbation of $P$**   The method in Section 5.3 introduces a variation of the log likelihood ratio test where the watermarked distribution $P_{M,w}$ is perturbed, resulting in a new optimization problem. Similarly, we could introduce a perturbation to the original distribution $P_M$. Specifically, we would adjust the original constraint of $\langle P_i, \exp(S_i) \rangle \leq 1$ to be $\langle P'_i, \exp(S_i) \rangle \leq 1, \forall P'_i, \text{s.t.} TV(P_i, P'_i) \leq d'$, where $TV(P_i, P'_i)$ denotes the total variation distance between $P_i$ and $P'_i$ and $d'$ is a small positive number.

This new optimization problem can be solved using similar methods as those in Appendix B.5.2. We have implemented this computation in our codebase. However, for the experiments in this paper, we only used the case where $d' = 0$.

# D    LIKELIHOOD-AGNOSTIC WATERMARK SCORE

Our unbiased watermark can also be detected in a likelihood-agnostic way such that it does not rely on a language model and its output logits to compute the score.

## D.1    METHOD

### D.1.1    REWEIGHTING FUNCTION

We use the same $\delta$-reweighting as in Section 4.1.2, but with a different implementation. Instead of using inverse sampling, we can also use Gumbel trick. Specifically, each watermark code is a list of $|\Sigma|$ number of independent and identically distributed standard Gumbel variables. The watermark code space is $\mathcal{E} = \mathbb{R}^\Sigma$. The probability density function of the watermark code is given by $P_E(E) = \prod_{a \in \Sigma} e^{-E(a) + e^{E(a)}}$.

To sample a token using the Gumbel trick, we compute $a^* = \operatorname{argmax}_a \{\log P(a) + E(a)\}$, and the reweighted distribution becomes $Q = \delta_{a^*}$. Gumbel variables allow us to guess the likelihood of a token coming from the watermark model without relying on logits, as tokens with larger Gumbel variables are more likely to be picked by the watermark model.

### D.1.2    SCORE DESIGN AND TAIL BOUND

Similar to Section 5, we calculate scores for each token, but without relying on the original and reweighted distribution $P$ and $Q$. Thus, the design of the likelihood-agnostic score has a certain degree of arbitrariness, unlike the method in Sections 5.2 and 5.3 which was derived in a principled way.

We choose the score to be $s_i = \ln 2 - \exp(-E(a^*))$. One of the main concerns of this construction is that it can yield a tail bound similar to Theorem 8.

**Theorem 16.** *For $n$ independent random variables $G_i \sim Gumbel(0, 1)$, if we define $s_i = \ln 2 - \exp(-G_i)$, we have $\mathbb{E}[\exp(s_i)] \leq 1$ and $P(\sum_{i=1}^n s_i \leq t) \leq e^{-t}$.*

For a token with watermark, the average score is $\mathbb{E}[\ln 2 - \exp(-G_i(a^*))] = \ln 2 - \sum_{a \in \Sigma} P(a)^2 = \ln 2 - \exp(-H_2(P))$, where $H_2(P)$ is the Rényi entropy of order 2. Therefore, the average score is positive only when the entropy is high.

Note that Theorem 16 requires independence of $s_i$, unlike Theorem 8 where the $s_i$ can be a random process. In practice, the Gumbel variables depend on the watermark code, and the watermark code might repeat, leading to dependencies between Gumbel variables and thus between scores. To address this issue, for repeating context codes, we set the score to zero, ensuring that Theorem 16 remains applicable.

The detection process is as follows: given a text $\boldsymbol{x}_{1:n} = (x_1, \ldots, x_n)$, we obtain a series of context codes $(cc_1, \ldots, cc_n)$ and watermark codes $(E_1, \ldots, E_n)$. The final scores are computed as

$$s_i = \begin{cases} \ln 2 - \exp(-E_i(x_i)) & \text{if } cc_i \notin cc_1, \ldots, cc_{i-1}, \\ 0 & \text{if } cc_i \in cc_1, \ldots, cc_{i-1}. \end{cases}$$

## D.2    COMPARISON BETWEEN LIKELIHOOD-BASED SCORE AND LIKELIHOOD-AGNOSTIC SCORE

Compared to the likelihood-based score, the likelihood-agnostic score has some notable drawbacks.

As it does not rely on logits, it cannot distinguish between high and low entropy situations. In low entropy cases, the likelihood-agnostic score still tends to have a large absolute value, even though it does not provide any signal and only contributes noise, lowering the score. In extreme cases, when the entropy is zero, the generation result is deterministic, and the ideal detection algorithm should output a zero score, as there is no evidence for or against the presence of the watermark. However, the likelihood-agnostic score would output a negative average score, giving a false indication that the text was not generated by a model with watermark.

Moreover, in cases where the original distribution $P_M$ is known, the likelihood-agnostic score is much smaller than the log likelihood ratio based score. According to the Neyman-Pearson lemma, the log likelihood ratio test is the most powerful statistical test, and its maximin variant also retains this property to a certain degree, thus providing a higher score than likelihood-agnostic score.

On the other hand, the likelihood-agnostic score has a lower computational cost, as it does not depend on the logits computed by a large language model. Furthermore, the fact that likelihood-agnostic score is independent of logits from the language model makes it more appealing when the original distribution $P_M$ is hard to estimate during detection.

## E  DETAILED EXPERIMENT SETUP

We evaluate the performance of our Unbiased Watermarks on two important applications of seq2seq models: text summarization(TS) and machine translation(MT).

**Text summarization.** In the TS task, we adopt the test set of of CNN-DM (Hermann et al., 2015) corpus, which consists of 11,490 examples. The model applied is BART-large, which contains 400 million parameters.

**Machine translation.** For the MT task, we employ the WMT'14 English (En) to Romanian (Ro) dataset, which has a test set size of 1,999 examples. The Multilingual Bart (MBart) (Liu et al., 2020) model and its official tokenizer is applied.

**Watermark setup.** We evaluate two reweighting functions in our experiment: $\delta$-reweight and $\gamma$-reweight. For context code generation, we employ the most recent five tokens as context code. For example, if the current input to the decoder is $(x_1, x_2, x_3)$, the context code used in generating $x_4$ would be $(x_1, x_2, x_3)$, considering only three tokens are available. Context code history is reset before generating each batch, thereby making our method $b$-shot-undetectable given a batch size of $b$. For the unbiased watermark code generation function, we use SHA-256 as the hash function and a 1024-bit random bitstring as the key $k$. The watermark code $E$ is sampled from $P_E$ using $\text{hash}(c, k)$ as the random seed.

In addition, we compared our method with the soft-red-list watermarking method from Kirchenbauer et al. (2023). Their method depends on two parameters $\delta$, controlling the size of the change in logits, and $\gamma$, which is the proportion of the green list in the total vocabulary. We test $\delta$ with three values: $0.0, 1.0, 2.0$, and fix $\gamma$ to be $\frac{1}{2}$. It is important to clarify that the $\delta$ and $\gamma$ in our $\delta$-reweight and $\gamma$-reweight are different from those in Kirchenbauer et al.'s method. In the latter, $\delta$ and $\gamma$ are hyperparameters, while in our method, $\delta$-reweight and $\gamma$-reweight are names of two reweighting strategies.

**Watermark detection.** We employ the maximin variant of LLR score for watermark detection. The score depends on a perturbation strength $d$ and is optimized by performing a grid search over the set $\{0, 0.1, \ldots, 0.9, 1.0\}$, which consists of 11 points. The optimal perturbation strength is the one that yields the highest score sum.

**Evaluation metrics.** For the TS task, we employ the ROUGE score (Lin, 2004), which measures the overlap in terms of n-grams to assess the effectiveness of the summary in capturing the essential content from the reference summaries. For the MT task, we use the BLEU score (Papineni et al., 2002) that emphasizes the lexical similarity between the machine-generated translations and the human reference translations. We estimated the distribution and standard error of BLEU score based on bootstrapping. In both tasks, we also apply BERTScore and Perplexity as auxiliary metrics.

**Computational costs.** Our experiments are carried out on a machine equipped with 2x AMD EPYC 7513 32-Core Processor and 8x A6000 GPUs. All experiments can be completed within 4 hours.

**Implementation.** The experiments are implemented based on the Huggingface library (Wolf et al., 2019), a popular platform for developing and sharing models in the NLP community.

# F  MORE EXPERIMENT

## F.1  ADDING WATERMARK

Tables 4 and 5 shows more result under the same setup as Table 1.

Table 4: Additional result about the performance of different watermarking methods on TS. We scale BERTScore and ROUGE with a factor of 100.

|  | BERTScore.Precision ↑ | BERTScore.Recall ↑ | ROUGE-2 ↑ | ROUGE-L ↑ |
|---|---|---|---|---|
| No Watermark | $0.3180 \pm 0.0009$ | $0.3361 \pm 0.0010$ | $0.1388 \pm 0.0008$ | $0.2445 \pm 0.0008$ |
| $\delta$-reweight | $0.3180 \pm 0.0009$ | $0.3365 \pm 0.0010$ | $0.1392 \pm 0.0008$ | $0.2451 \pm 0.0008$ |
| $\gamma$-reweight | $0.3180 \pm 0.0009$ | $0.3360 \pm 0.0010$ | $0.1397 \pm 0.0008$ | $0.2451 \pm 0.0008$ |
| Soft($\delta$=0.0) | $0.3180 \pm 0.0009$ | $0.3361 \pm 0.0010$ | $0.1388 \pm 0.0008$ | $0.2445 \pm 0.0008$ |
| Soft($\delta$=1.0) | $0.3092 \pm 0.0009$ | $0.3382 \pm 0.0009$ | $0.1344 \pm 0.0007$ | $0.2400 \pm 0.0007$ |
| Soft($\delta$=2.0) | $0.2908 \pm 0.0008$ | $0.3339 \pm 0.0009$ | $0.1238 \pm 0.0007$ | $0.2293 \pm 0.0007$ |

GPTScore (Fu et al., 2023) is an LLM based auto evaluator. We utilize text-curie-001 for our evaluations.

Table 5: Additional result about the performance of different watermarking methods on MT. We scale BERTScore with a factor of 100.

|  | BERTScore.Precision ↑ | BERTScore.Recall ↑ | Perplexity ↓ | GPTScore ↓ |
|---|---|---|---|---|
| No Watermark | $0.546 \pm 0.003$ | $0.575 \pm 0.003$ | $2.31 \pm 0.07$ | $1.26 \pm 0.01$ |
| $\delta$-reweight | $0.550 \pm 0.003$ | $0.579 \pm 0.003$ | $2.20 \pm 0.05$ | $1.25 \pm 0.01$ |
| $\gamma$-reweight | $0.549 \pm 0.003$ | $0.577 \pm 0.003$ | $2.24 \pm 0.04$ | $1.26 \pm 0.01$ |
| Soft($\delta$=0.0) | $0.546 \pm 0.003$ | $0.575 \pm 0.003$ | $2.31 \pm 0.07$ | $1.26 \pm 0.01$ |
| Soft($\delta$=1.0) | $0.537 \pm 0.003$ | $0.568 \pm 0.003$ | $2.43 \pm 0.07$ | $1.31 \pm 0.01$ |
| Soft($\delta$=2.0) | $0.523 \pm 0.003$ | $0.555 \pm 0.003$ | $2.81 \pm 0.07$ | $1.41 \pm 0.01$ |

## F.2  SENSITIVITY OF SCORES

The detection methods in Sections 5.2 and 5.3 rely on the output logits of the language models, which in turn depend on various factors such as the prompt, the temperature and sampling policy used during the generation process, and the language model itself. In this section, we measure the sensitivity of the scores to changes in these parameters.

Watermarked samples are generated from the distribution $P_{M,w}$, which comes from reweighting of the original distribution $P_M$. However, during detection, we modify some parameters, including temperature, sampling policy (top_k), input, and model, resulting in a new probability distribution $P'_M$.

The following table demonstrates the decrease in scores under different changes, showing that when $P'_M$ is not equal to $P_M$, the scores decline. This implies that more tokens are required to accumulate sufficient evidence to prove the existence of the watermark.

Table 6: Score per token when the estimated token distribution is computed from a different temperature than the real token distribution.

|  | Text summarization | | Machine translation | |
|---|---|---|---|---|
| temperature | $\delta$-reweight | $\gamma$-reweight | $\delta$-reweight | $\gamma$-reweight |
| 0.5 | $0.049 \pm 0.407$ | $0.133 \pm 0.309$ | $0.041 \pm 0.303$ | $0.084 \pm 0.241$ |
| 1.0 (groundtruth) | $0.878 \pm 1.435$ | $0.220 \pm 0.367$ | $0.420 \pm 1.135$ | $0.105 \pm 0.291$ |
| 1.5 | $0.036 \pm 0.498$ | $0.166 \pm 0.455$ | $0.019 \pm 0.324$ | $0.088 \pm 0.335$ |

Comparing the two reweight functions, we find that when $P'_M$ is equal to $P_M$, the $\delta$-reweight always yields a higher score than the $\gamma$-reweight. However, when $P'_M$ is different from $P_M$, the scores

Table 7: Score per token when the estimated token distribution is computed from a different top_k than the real token distribution.

| top_k | Text summarization | | Machine translation | |
|---|---|---|---|---|
| | $\delta$-reweight | $\gamma$-reweight | $\delta$-reweight | $\gamma$-reweight |
| 20 | $0.520 \pm 1.144$ | $0.212 \pm 0.362$ | $0.274 \pm 0.859$ | $0.101 \pm 0.284$ |
| 50 (groundtruth) | $0.878 \pm 1.435$ | $0.220 \pm 0.367$ | $0.420 \pm 1.135$ | $0.105 \pm 0.291$ |
| 100 | $0.582 \pm 1.262$ | $0.219 \pm 0.369$ | $0.288 \pm 0.930$ | $0.105 \pm 0.292$ |
| No top_k sampling | $0.377 \pm 1.124$ | $0.216 \pm 0.373$ | $0.022 \pm 0.349$ | $0.097 \pm 0.324$ |

Table 8: Score per token when the estimated token distribution is computed with and without input.

| | Text summarization | | Machine translation | |
|---|---|---|---|---|
| | $\delta$-reweight | $\gamma$-reweight | $\delta$-reweight | $\gamma$-reweight |
| with input (groundtruth) | $0.8783 \pm 1.4353$ | $0.2206 \pm 0.3677$ | $0.4201 \pm 1.1355$ | $0.1058 \pm 0.2916$ |
| without input | $0.0108 \pm 0.2170$ | $0.0244 \pm 0.2417$ | $0.0096 \pm 0.2004$ | $0.0186 \pm 0.1904$ |

Table 9: Score per token when the estimated token distribution is computed from a different model than the real token distribution.

| model | Text summarization | |
|---|---|---|
| | $\delta$-reweight | $\gamma$-reweight |
| "philschmid/bart-large-cnn-samsum" (groundtruth) | $0.878 \pm 1.435$ | $0.220 \pm 0.367$ |
| "facebook/bart-large-cnn" | $0.041 \pm 0.447$ | $0.091 \pm 0.412$ |

obtained from the $\delta$-reweight exhibit a significant drop, whereas the decline in scores for the $\gamma$-reweight is always more gradual than that of the $\delta$-reweight. This indicates that the $\gamma$-reweight is less sensitive to the differences between $P'_M$ and $P_M$.

### F.3 LIKELIHOOD-AGNOSTIC SCORE

When applied to text summarization, which is a task with relatively high entropy, the likelihood-agnostic score is positive on average but an order of magnitude lower than the likelihood-based score. For machine translation, which is a low entropy task, the average score is negative, and thus cannot be used to detect watermark in this case.

Table 10: Mean and variance of score per token for $\delta$-reweight based on Gumbel trick on different tasks.

| | Text summarization | Machine translation |
|---|---|---|
| Maximin variant of LLR score | $0.876 \pm 1.444$ | $0.429 \pm 1.172$ |
| Likelihood-agnostic score | $0.078 \pm 0.776$ | $-0.104 \pm 0.891$ |

### F.4 VERIFYING DOWNSTREAM-INVARIANT PROPERTY OF WATERMARK FOR MORE MODELS

Table 11: Additional result with T5 for translation tasks and LLaMA 2 (Touvron et al., 2023) for summarization and poem generation.

| Task | Text summarization | Machine translation | Poetry generation |
|---|---|---|---|
| Model | Llama 2 | T5 | Llama2 |
| Score | ROUGE-1 $\uparrow$ | BERTScore.Precision $\uparrow$ | Perplexity $\downarrow$ |
| No Watermark | $0.3705 \pm 0.0009$ | $0.575 \pm 0.003$ | $2.73 \pm 0.08$ |
| $\delta$-reweight | $0.3704 \pm 0.0009$ | $0.577 \pm 0.003$ | $2.71 \pm 0.06$ |
| $\gamma$-reweight | $0.3704 \pm 0.0009$ | $0.576 \pm 0.003$ | $2.71 \pm 0.08$ |
| Soft($\delta$=0.0) | $0.3705 \pm 0.0009$ | $0.575 \pm 0.003$ | $2.73 \pm 0.08$ |
| Soft($\delta$=1.0) | $0.3678 \pm 0.0009$ | $0.571 \pm 0.003$ | $3.04 \pm 0.13$ |
| Soft($\delta$=2.0) | $0.3610 \pm 0.0009$ | $0.560 \pm 0.003$ | $3.92 \pm 0.16$ |

### F.5 ROBUSTNESS OF WATERMARKS

In this section, we aim to evaluate the robustness of watermarking methods. To perform this assessment, we first initialize 512 string prompts for open-ended text completion. For each of these prompt, we use certain watermark method to generate 16 tokens sequentially. These generated strings are modified and then analyzed to detect the presence of watermarks.

In order to test the resilience of the watermarks against noise and alterations, we introduce random perturbation to generated text by replacing a $\varepsilon$ portion of the tokens with random tokens. We start our experiment with $\varepsilon = 0.0$, indicating no perturbation to the original strings, and gradually increase it to $\varepsilon = 0.5$, where half of the tokens in each string are replaced with random tokens.

To quantify the robustness of the watermarks, we calculate a corresponding score for each level of perturbation and measure the Area Under the Curve (AUC). For unbiased watermark methods, the score is calculated using the method described in section Section 5.3. For soft red list methods, we employ the z-score as defined in Kirchenbauer et al. (2023).

Table 12: AUC for different watermarking detection methods under different perturbation strength.

| | $\delta$-reweight | $\gamma$-reweight | Soft($\delta = 1.0$) | Soft($\delta = 2.0$) |
|---|---|---|---|---|
| $\epsilon = 0.0$ | $0.9997 \pm 0.0005$ | $0.9936 \pm 0.0016$ | $0.8446 \pm 0.0069$ | $0.9705 \pm 0.0030$ |
| $\epsilon = 0.1$ | $0.9569 \pm 0.0021$ | $0.9297 \pm 0.0030$ | $0.7871 \pm 0.0081$ | $0.9239 \pm 0.0070$ |
| $\epsilon = 0.2$ | $0.8881 \pm 0.0043$ | $0.8391 \pm 0.0018$ | $0.7339 \pm 0.0110$ | $0.8680 \pm 0.0088$ |
| $\epsilon = 0.3$ | $0.8152 \pm 0.0059$ | $0.7574 \pm 0.0054$ | $0.6741 \pm 0.0119$ | $0.7956 \pm 0.0110$ |
| $\epsilon = 0.4$ | $0.7487 \pm 0.0056$ | $0.6942 \pm 0.0107$ | $0.6334 \pm 0.0084$ | $0.7312 \pm 0.0121$ |
| $\epsilon = 0.5$ | $0.6851 \pm 0.0067$ | $0.6502 \pm 0.0068$ | $0.5859 \pm 0.0079$ | $0.6561 \pm 0.0124$ |

Futhermore, we supplemented our research with an evaluation against a random deletion attack, where about $\epsilon$ portion of tokens are deleted:

Table 13: AUC for different watermarking detection methods under different deletion portion.

| | $\delta$-reweight | $\gamma$-reweight | Soft($\delta = 1.0$) | Soft($\delta = 2.0$) |
|---|---|---|---|---|
| $\epsilon = 0.0$ | $0.9997 \pm 0.0005$ | $0.9936 \pm 0.0016$ | $0.8446 \pm 0.0069$ | $0.9705 \pm 0.0030$ |
| $\epsilon = 0.1$ | $0.9674 \pm 0.0026$ | $0.9348 \pm 0.0045$ | $0.7895 \pm 0.0044$ | $0.9257 \pm 0.0030$ |
| $\epsilon = 0.2$ | $0.9107 \pm 0.0061$ | $0.8616 \pm 0.0032$ | $0.7377 \pm 0.0079$ | $0.8697 \pm 0.0043$ |
| $\epsilon = 0.3$ | $0.8319 \pm 0.0056$ | $0.7697 \pm 0.0023$ | $0.6903 \pm 0.0074$ | $0.8042 \pm 0.0067$ |
| $\epsilon = 0.4$ | $0.7634 \pm 0.0011$ | $0.7020 \pm 0.0054$ | $0.6387 \pm 0.0026$ | $0.7336 \pm 0.0084$ |
| $\epsilon = 0.5$ | $0.7032 \pm 0.0044$ | $0.6577 \pm 0.0082$ | $0.5894 \pm 0.0075$ | $0.6626 \pm 0.0104$ |

## G LIMITATIONS

### G.1 MAJOR LIMITATIONS

We note that our unbiased watermarking technique only works for generative processes with high entropy. In an extreme case, when entropy is 0 and output of the original model is fixed, any unbiased watermarking method will always yield the same result as the original model. As a result, it is challenging to integrate our unbiased watermarking approach with beam search algorithms due to their intrinsic deterministic nature.

### G.2 MINOR LIMITATIONS

- Since our study is focused on unbiasedness, rather than robustness of watermark method, we only test the robustness of the watermark for a single attacks, that is the random substitution attack. There are numerous ways of watermark removal ranging from simple text insertion to more sophisticated methods like paraphrasing attacks. These attacks have their own implication on watermark robustness, but this topics is beyond the scope of this paper.
- Even though we have proposed a watermarking framework, there is considerable design space left unexplored. Many reweighting functions and context codes may be applicable, but it is unclear

which one is optimal in practice, particularly since we currently lack standard evaluation metrics. We expect that continued research in this area could possibly shed more light on this subject.

- In Algorithm 1, the introduction of context code history strictly ensures n-shot-undetectable watermarking at the expense of additional storage requirements, as the context code history from past generation processes needs to be retained. This presents a trade-off between storage and undetectability. For instance, if we store all context codes in the previous $n$ generated outputs, we can ensure $n$-shot-undetectability. However, the greater the value of $n$, the larger the required storage space, though this does provide stronger undetectability. Generally, storage complexity increases with $O(n)$.

