# OpenReview forum: "Unbiased Watermark for Large Language Models"
_ICLR.cc/2024/Conference — ICLR 2024 spotlight_

### Official Review · Reviewer_MH5n · 2023-10-30

**Soundness:** 3 good
**Presentation:** 3 good
**Contribution:** 3 good
**Rating:** 6
**Confidence:** 3

**Summary:**

The paper explores the use of unbiased watermarks for large language models (LLMs) to track and attribute model outputs without compromising output quality. The authors introduce an innovative family of watermark methods that guarantee non-degradation of text quality and offer a comprehensive framework for designing and detecting unbiased watermarks.

**Strengths:**

1. Improved accountability: Unbiased watermarks can help track and attribute model outputs to specific providers, which can improve accountability and transparency in the use of LLMs. This can be particularly important in applications such as automated content generation, where it may be difficult to determine the source of generated content.

2. Maintaining output quality: Previous studies have suggested that watermarking can compromise the quality of model-generated outputs. However, the use of unbiased watermarks can maintain output quality while still providing a means of tracking and attribution.

3. Protection against misuse: The use of unbiased watermarks can help protect against the potential misuse of LLMs, such as the generation of fake news or other malicious content. This can help ensure that LLMs are used responsibly and ethically.

4. Flexibility: The authors offer a comprehensive framework for designing and detecting unbiased watermarks, which can provide flexibility in the implementation of watermarking for different applications and use cases. This can help ensure that unbiased watermarks are tailored to specific needs and requirements.

5. Generality: The authors propose a mathematically well-defined framework for watermark trade-off.

**Weaknesses:**

1. Potential limitations: The paper does not address the potential limitations or challenges of implementing unbiased watermarks in practice. It is possible that there may be technical or logistical challenges that could make it difficult to implement unbiased watermarks effectively.

2. Complexity: The implementation of unbiased watermarks may require significant technical expertise and resources, which could be a barrier to adoption for some providers. Additionally, the complexity of the watermarking process could potentially impact the performance of LLMs or other downstream tasks.

**Questions:**

How do you envision the implementation of unbiased watermarks in practice, and what challenges do you anticipate?

How do you plan to address concerns (in practice) regarding the potential manipulation or removal of watermarks by users?

Can you elaborate the key factors when chasing the optimal trade-off between mentioned in the paper?

---

> ### Author Response · Authors · 2023-11-16
> **Response to Reviewer MH5n**
>
> Thank you for acknowledging our work on the unbiased watermarks. We appreciate your thoughtful comment and would like to answer the questions point by point.
>
> > How do you envision the implementation of unbiased watermarks in practice, and what challenges do you anticipate?
>
> Regarding the practical implementation of unbiased watermarks, storage issue could be a consideration. In our algorithm, we introduced the concept of context code history, which inevitably requires additional storage space. While theoretically, unlimited storage could enable $\infty$-shot-undetectable watermarking, practical constraints might limit storage capacity, resulting in n-shot-undetectability. As n increases, so does the required storage space, but this trade-off improves undetectability. For instance, storing the context code of the last n outputs guarantees n-shot-undetectability. Generally, storage complexity grows in $O(n)$. More details can be found in Appendix G.3, item three.
>
> > How do you plan to address concerns (in practice) regarding the potential manipulation or removal of watermarks by users?
>
> We believe the battle between watermarking and watermark removal will be a long-living problem. This dynamic has seen rapid developments recently. Our current research marks significant progress in this area by theoretically ensuring that users cannot detect the presence of watermarks. This advancement complicates the manipulation and removal of watermarks, as it becomes challenging to assess whether the watermark remains intact. We look forward to future research that will build upon these advancements.
>
> > Can you elaborate the key factors when chasing the optimal trade-off between mentioned in the paper?
>
> One of our core contributions is the decoupling of unbiasedness and watermark strength. Previous works faced a trade-off between output quality and watermark detectability; to make watermarks easier to detect, one had to sacrifice quality. Our approach ensures that output quality remains unchanged, regardless of watermark strength. Thus, we believe we have achieved an optimal trade-off between these factors.
>
> Again, thank you for your insightful questions and the opportunity to clarify our approach.

---

> > ### Comment · Reviewer_MH5n · 2023-11-21
> >
> > Thanks for the authors' response, I keep my score unchanged.

---

### Official Review · Reviewer_7gzS · 2023-10-31

**Soundness:** 3 good
**Presentation:** 3 good
**Contribution:** 4 excellent
**Rating:** 8
**Confidence:** 4

**Summary:**

The paper introduces the "unbiased watermarks" in large language models, challenging the prevalent belief that watermark strength negatively affects output quality. Two novel watermarking techniques, $\delta$-reweight, and $\gamma$-reweight, are proposed; they maintain output quality in machine translation and text summarization tasks. Through theoretical analysis, they proved two methods could guarantee the unbiasedness of the watermarked generation. Additionally, a new log-likelihood ratio test is presented for watermark detection, which has theoretical guarantees on type I errors.

**Strengths:**

1. The concept of an "unbiased watermark" is interesting and crucial for practical implementation in real-world scenarios.

2. The paper introduces two novel methodologies that are not only straightforward (simple logit reweighting) , but also empirically effective, as demonstrated through experimental evaluation.

3. The study proposes a novel maximin variant of the Log-Likelihood Ratio (LLR) score for hypothesis testing, exhibiting enhanced robustness compared to the  LLR score.

**Weaknesses:**

My major concern about this paper is the lack of comprehensive evaluations:
1. The evaluations do not contain any attacks (the authors mentioned them in Appendix A2). I highly recommend the authors evaluate those attacks. If the proposed watermark methods suffer from a simple attack like paraphrasing, it would significantly undermine the paper's contributions.
2. Larger models like LLAMA-2 should also be evaluated to further enhance the practicality.

**Questions:**

NA

---

> ### Author Response · Authors · 2023-11-16
> **Response to Reviewer 7gzS**
>
> Thank you for recognizing the strengths of our work and providing constructive feedback.
> We would like to address the concerns highlighted in your review:
>
> > The evaluations do not contain any attacks (the authors mentioned them in Appendix A2)
>
> Actually, we have included preliminary evaluations against random substitution attack in Appendix F.5. Additionally, we have now supplemented our research with an evaluation against a random deletion attack:
>
> |                 |     0.0      |     0.1      |     0.2      |     0.3      |     0.4      |     0.5      |
> |-----------------|-------------|-------------|-------------|-------------|-------------|-------------|
> | Delta_Reweight  | 0.9997±0.0005 | 0.9674±0.0026 | 0.9107±0.0061 | 0.8319±0.0056 | 0.7634±0.0011 | 0.7032±0.0044 |
> | Gamma_Reweight  | 0.9936±0.0016 | 0.9348±0.0045 | 0.8616±0.0032 | 0.7697±0.0023 | 0.7020±0.0054 | 0.6577±0.0082 |
> | Soft(delta=1.0) | 0.8446±0.0069 | 0.7895±0.0044 | 0.7377±0.0079 | 0.6903±0.0074 | 0.6387±0.0026 | 0.5894±0.0075 |
> | Soft(delta=2.0) | 0.9705±0.0030 | 0.9257±0.0030 | 0.8697±0.0043 | 0.8042±0.0067 | 0.7336±0.0084 | 0.6626±0.0104 |
>
> > If the proposed watermark methods suffer from a simple attack like paraphrasing, it would significantly undermine the paper's contributions.
>
> Unfortunately, it is true that our watermarking methods are vulnerable to simple attacks. Specifically, although the watermarks can still be detected after changes to the input, the efficiency of this detection drops notably, necessitating longer sequences for reliable watermark detection. This vulnerability is fundamentally due to the fact that watermark detection relies on the presence of identical n-grams during watermark generation and detection. Any form of attack, including paraphrasing or simple word substitutions, could easily disrupt these n-grams, thereby reducing the efficiency of watermarks detection. This vulnerability is not unique to our work; it is a common challenge in the field of watermarking, similar to recent findings (Kirchenbauer et al. 2023).
>
> The primary contribution of our paper is the development of the concept of an unbiased watermark. While we recognize the importance of robustness in watermarking research, our current work is not aimed at enhancing the robustness of watermarks. Our approach maintains a level of robustness on par with existing methodologies, as discussed in Appendix F.5.
>
> > Larger models like LLAMA-2 should also be evaluated to further enhance the practicality.
>
> We have conducted evaluations with larger models, including LLAMA-2 in our appendix. The results of these evaluations are detailed in Appendix F.1, where we used LLAMA-2 for text summarization and poetry generation tasks. Furthermore, we have expanded our experiments to include natural language generation with Wikitext and instruction following with FLAN, providing additional practical evidence for downstream invariant property:
>
> | Model                       | Wikitext (Llama 2), PPL | Flan (Llama 2), PPL |
> |-----------------------------|------------------------|---------------------|
> | $\delta$-reweight           | 1.692±0.009                       | 2.57±0.03           |
> | $\gamma$-reweight           | 1.719±0.011                       | 2.557±0.028         |
> | Soft($\delta$=0.0)          | 1.693±0.011                       | 2.558±0.025         |
> | Soft($\delta$=1.0)          | 1.761±0.011                       | 2.697±0.028         |
> | Soft($\delta$=2.0)          | 2.025±0.017                       | 3.17±0.04           |
>
> Thank you again for your thoughtful comment. Your recognition of our work's strengths and constructive feedback are greatly appreciated.

---

> > ### Comment · Reviewer_7gzS · 2023-11-20
> >
> > Thank the authors for a detailed reply. I hope those experiments can be incorporated into the paper.
> >
> > Overall, I vote "accept" for the paper for its great contribution to the "unbiased" LLM watermark.

---

### Official Review · Reviewer_usqd · 2023-11-01

**Soundness:** 3 good
**Presentation:** 3 good
**Contribution:** 3 good
**Rating:** 6
**Confidence:** 1

**Summary:**

The paper studied watermarking LLM with two goals: a) is efficiently detectable by the service provider; b) can’t be detected by users and does not negatively impact the quality of the output. The experiments are comprehensive that they also explore that output quality can be well preserved in downstream tasks, such as machine translation and text summarization.

**Strengths:**

The proposed two reweighing algorithms are backed by solid mathematical foundations.
The experiments are quite comprehensive, and I enjoyed reading about them.
The emphasis on unbiasedness and downstream tasks adds significant value to the paper.

**Weaknesses:**

As someone who is not very familiar with this topic, I find the paper to be well-organized, solid, and supported by comprehensive experiments. I do not see any obvious weaknesses in the paper from my perspective.

**Questions:**

I am confused by the `NW' output in Table 3. It appears to be quite unreadable, especially when compared to the outputs with watermarking strategies. Is this because the OPT-6.7B model is not fine-tuned for the NW cases? I am concerned about whether this comparison is fair enough.

---

> ### Author Response · Authors · 2023-11-16
> **Response to Reviewer usqd**
>
> Thanks for recognizing our novel contribution.
>
> > Is this because the OPT-6.7B model is not fine-tuned for the NW cases? I am concerned about whether this comparison is fair enough.
>
> We didn't finetune the model, and we use the same model OPT-6.7B across NW, $\delta$-RW, $\gamma$-RW to ensure a fair comparison.
>
> The lower readability output in the 'NW' section is due to the inherent variability of the OPT-6.7B model's performance. This model sometimes generates lower-quality outputs, which is evident in the NW case. This is a characteristic of the OPT-6.7B model itself, rather than an effect of our watermarking strategy or a lack of fine-tuning.
>
> In light of your feedback, we have replicated table 3 with a state-of-the-art model, LLaMA2. We find that the readability of the outputs improve significantly, and the variability of output quality also decreases. The updated results with LLaMA2 are as follows:
>
> | Prompt   | What is a watermark? What’s the purpose of it? | score | p\-value |
> |----|----|----|----|
> |NW | What are the different types of watermarks?<br><br>A watermark is a transparent image or pattern embedded in a digital image, document or video, that can be seen when the image is enlarged or magnified. The purpose of a watermark is to provide a means of identifying the owner of the image, document or video, or to indicate that the image has been altered or copied without| 0.0 | 11|
> | $\delta$-RW | What are the benefits of using a watermark? Are there different types of watermarks? How do I create a watermark for my images?<br><br>What is a watermark?<br><br>A watermark is a visual representation of a brand, logo, or text that is embedded or superimposed over an image or document. The purpose of a watermark is | 15.24 | 2.6e-6 |
> | $\gamma$-RW | What are the different types of watermarks? What are the features of a digital watermarking system? Who are the parties involved in a digital watermarking process? What are the applications of digital watermarking? What are the challenges in digital watermarking? What is the future of digital watermarking? Conclusion Questions and Answers References and Recommendations | 8.86 | 0.00155 |
>
> We hope that the additional information will address your concerns. Once again, thank you for recognizing our work and providing valuable feedback.

---

### Official Review · Reviewer_ZzM6 · 2023-11-04

**Soundness:** 4 excellent
**Presentation:** 3 good
**Contribution:** 4 excellent
**Rating:** 8
**Confidence:** 4

**Summary:**

In this paper, the authors propose a unbiased watermarking method for LLM generated text, aiming at addressing the problem of attribution the model outputs without compromising the output quality. Specifically, two distribution re-weighting strategies are proposed to satisfy the unbiased property, and the corresponding score-based hypothesis testings are proposed. Furthermore, the authors evaluate the effectiveness of the proposed method on summarization and translation tasks.

**Strengths:**

1. The authors propsoe two unbiased reweighting methods and the theoretical proofs are given.

2. The authors propose a novel maximin variant of LLR score, which is more robust than previous LLR score

3. The authors conduct the experiment on machine translation and text summarization tasks, showing the quality of the generated text.

**Weaknesses:**

1. The tasks included in the experiment are only summarization and translation, and more tasks, such as general natural language generation, are required to demonstrate the effectiveness of the proposed method.


2. It seems that the robustness towards the exiting watermarking attacks are not verified in the paper. It seems that only in Appendix F.5, the authors evaluate the robustness under random perturbation. More attacks mentioned in Appendix A.2 are expected.


3. Minor: Since the highlight of the proposed method is the unbiased property, in my opinion, the theorems 14 should be in the main body instead of the appendix.

**Questions:**

See weakness 1 & 2.

---

> ### Author Response · Authors · 2023-11-16
> **Response to Reviewer ZzM6**
>
> Thank you for your comprehensive review and constructive feedback on our paper.
>
> > such as general natural language generation, are required
>
> We appreciate your suggestion to extend our experiments beyond translation and summarization tasks. We have expanded our datasets to include wikitext for natural language completion and the FLAN dataset for instruction-following tasks. Our results demonstrate the downstream-invariant properties across these new tasks.
>
> | Model                       | Wikitext (Llama 2), PPL | Flan (Llama 2), PPL |
> |-----------------------------|------------------------|---------------------|
> | $\delta$-reweight           | 1.692±0.009                       | 2.57±0.03           |
> | $\gamma$-reweight           | 1.719±0.011                       | 2.557±0.028         |
> | Soft($\delta$=0.0)          | 1.693±0.011                       | 2.558±0.025         |
> | Soft($\delta$=1.0)          | 1.761±0.011                       | 2.697±0.028         |
> | Soft($\delta$=2.0)          | 2.025±0.017                       | 3.17±0.04           |
>
> > It seems that the robustness towards the exiting watermarking attacks are not verified in the paper
>
>
> Your point about the robustness of our watermarking method was well-taken. To address this, we've now incorporated tests against random token deletion. These additional experiments showcase the robustness of our methods. Exploring a broader range of sophisticated watermarking attack types, including emoji attack, will be a focus of our future work.
>
> |                 |     0.0      |     0.1      |     0.2      |     0.3      |     0.4      |     0.5      |
> |-----------------|-------------|-------------|-------------|-------------|-------------|-------------|
> | Delta_Reweight  | 0.9997±0.0005 | 0.9674±0.0026 | 0.9107±0.0061 | 0.8319±0.0056 | 0.7634±0.0011 | 0.7032±0.0044 |
> | Gamma_Reweight  | 0.9936±0.0016 | 0.9348±0.0045 | 0.8616±0.0032 | 0.7697±0.0023 | 0.7020±0.0054 | 0.6577±0.0082 |
> | Soft(delta=1.0) | 0.8446±0.0069 | 0.7895±0.0044 | 0.7377±0.0079 | 0.6903±0.0074 | 0.6387±0.0026 | 0.5894±0.0075 |
> | Soft(delta=2.0) | 0.9705±0.0030 | 0.9257±0.0030 | 0.8697±0.0043 | 0.8042±0.0067 | 0.7336±0.0084 | 0.6626±0.0104 |
>
> > Theorems 14 should be in the main body instead of the appendix
>
> Although we briefly mention that "Both methods are unbiased1" on page 4, we agree that a formal presentation of Theorem 14 in the main body would better clarify the argument. Accordingly, we will revise our manuscript to include Theorem 14 in a more prominent section.
>
> We're grateful for your constructive feedback and for your recognition of our novel contributions.

---

> > ### Comment · Reviewer_ZzM6 · 2023-11-22
> >
> > Thank you for your additional experiments. I will keep my score.

---

### Official Review · Reviewer_tWzc · 2023-11-04

**Soundness:** 2 fair
**Presentation:** 3 good
**Contribution:** 2 fair
**Rating:** 5
**Confidence:** 4

**Summary:**

The paper proposes two new algorithms for watermarking the outputs from language models, one based on gamma-reweight and other based on a delta-reweight. The algorithms are sound and non-distortionary, and can be used to watermark LLMs without provably reducing their utility. The authors present minimal experimental evidence illustrating the detectability of the watermarks on translation and summarization tasks, and further some quality measurements to demonstrate the quality of the LLM on the task remains unaffected from watermarking.

The "signature" being watermarked is based on the previously seen context, and they store all contexts so you do not watermark when you encounter the same context again. This is key to preserving the non-distortionary nature of the watermarks.

**Strengths:**

Strengths:
1. The formulation of the LLR approach is very interesting, and is novel to the best of my knowledge. It deviates from z-score based tests used in prior work, and seems promising (with certain caveats, see weaknesses).
2. While the gamma scheme overlap with the schemes presented in Kuditipudi et al. (I recognize the distinction between hashing and the sequence of keys protocols, there are strong similarities in the way the signature is encoded however), I acknowledge that it is concurrent work and has not been published yet at a peer reviewed venue. Hence, I would regard this as a novel contribution, and both the schemes are interesting and sound watermarking schemes.
3. This is an important and timely topic, and the algorithms presented are non-distortionary, and are a step forward in watermarks that can be adopted in the real-world without reducing the utility of the LLMs.

**Weaknesses:**

Cons:
1. The experiments are somewhat on the weaker side. I do not know how these schemes compare in relation to the UMD scheme in terms of detectability, or perform more generally (even if we disregard the UMD scheme since it is non-distortionary). Table 1 compares the two schemes in terms of quality of the outputs, but there is a lack of discussion of the watermarking performance more generally. Table 2 presents the evidence at a token level, but it would be much more interesting to give our some sort of a global metric, such as the AUC (or AUC-PR) in detecting watermarked vs unwatermarked text.

2. While the LLR scheme is novel and interesting, it is not clear how robust the maximin LLR scheme is to different scenarios. Meaningful ablation experiments are missing.

3. The authors proposed storing a log of all the responses so far. At some point, you will exhaust all the possible n-grams, and you will not be watermarking any more, and the watermark will get weaker and weaker over time. This seems like a practical limitation.

**Questions:**

1. How does the watermarking scheme perform on context completion and instruction following tasks? Is it possible to present some empirical evidence on how many tokens are needed to achieve a certain level of detectability on open source LLMs?

2. What happens if you do not have access to the original LLM? (e.g., if you watermarked with ChatGPT but ran verification with Llama-1b, the shift in distributions would be quite large). Would the maximin LLR scheme outperform a z-score based statistical test?

Even for the same model, if you have edits (e.g., you don't have access to the prompt, or if someone has shuffled the sentences in the text), does the maximin LLR scheme work?

---

> ### Author Response · Authors · 2023-11-16
> **Response to Reviewer tWzc**
>
> We appreciate the time and effort you have dedicated to reviewing our manuscript. We are grateful for the opportunity to address the concerns and queries you have brought forward.
>
> > While the gamma scheme overlap with the schemes presented in Kuditipudi et al. (... there are strong similarities in the way the signature is encoded however)
>
> We respectfully disagree with the assertion of strong similarities in signature encoding between our gamma reweight scheme and method in Kuditipudi et al.
>
> The critical distinction lies in the underlying mechanism: all schemes $\Gamma(\xi,\mu)$ presented in Kuditipudi et al. are based on inverse sampling, where the output is a specific value. In contrast, our gamma reweight scheme is not based on inverse sampling, as it produces a distribution rather than a single value。
>
> Given this fundamental difference, we do not concur with the notion of strong similarity. We would appreciate it if the reviewer could clarify what specific similarities are being referred to. This clarification will enable us to address the concern more directly and ensure that our method's uniqueness and contributions are accurately understood.
>
> > but it would be much more interesting to give our some sort of a global metric, such as the AUC (or AUC-PR) in detecting watermarked vs unwatermarked text
>
> Actually, we have reported AUC performance in Section F.5. In standard experiments, where no perturbations are introduced, the AUC is extremely close to 1, and the deviation from 1 is so minimal that it falls within the margin of error, making quantitative assessment challenging. This is the primary reason why we have opted to present the score per token in the main paper, as it provides a more discernible measure of the watermark's detectability.
>
> >  it is not clear how robust the maximin LLR scheme is to different scenarios. Meaningful ablation experiments are missing.
>
> We kindly direct your attention to Section F.2 in the appendix, where we provided ablation experiments specifically to explore the robustness of the maximin LLR approach in different scenarios. These experiments cover variations in temperature settings, top-k selections, different prompts, and different language models.
>
> We believe that these experiments meaningfully address concerns regarding the robustness of our approach.
>
> > you will exhaust all the possible n-grams
>
> While that is possible in theory, the number of all possible n-grams is astronomically large for $n\geq 3$, and in reality, it's nearly impossible to reaching saturation.
>
> Furthermore, refraining watermarking on previously seen n-grams is necessary for ensuring strict unbiased property. As such, we consider being the first to clearly explain this critical point as a significant contribution of our work, rather than a weakness.
>
> Additionally, if needed, there are straightforward modifications that can slightly relax the strict $\infty$-shot-undetectable property to circumvent any practical limitations, as discussed in Section G.2, third point, of the appendix.
>
> > How does the watermarking scheme perform on context completion and instruction following tasks?
>
> We've broadened our datasets to include wikitext for context completion and flan for instruction following. The downstream-invariant property is also observed in these new tasks.
>
> | Model                       | Wikitext (Llama 2), PPL | Flan (Llama 2), PPL |
> |-----------------------------|------------------------|---------------------|
> | $\delta$-reweight           | 1.692±0.009                       | 2.57±0.03           |
> | $\gamma$-reweight           | 1.719±0.011                       | 2.557±0.028         |
> | Soft($\delta$=0.0)          | 1.693±0.011                       | 2.558±0.025         |
> | Soft($\delta$=1.0)          | 1.761±0.011                       | 2.697±0.028         |
> | Soft($\delta$=2.0)          | 2.025±0.017                       | 3.17±0.04           |
>
> We want to emphasize that our methods are supported by rigorous theoretical guarantee, and the downstream-invariant property is independent of specific task.
>
> > Is it possible to present some empirical evidence on how many tokens are needed to achieve a certain level of detectability on open source LLMs?
>
> Yes. This can be computed by the score per token. For instance, if each token increases the score on average by 0.8784 (as reported in Table 2), then 12 tokens would be enough to achieve a score of 10.5, which means watermark can be detected with p-value < 0.0005, while 23 tokens would bring it down to p-value < 3e-8.

---

> > ### Author Response · Authors · 2023-11-16
> > **Response to Reviewer tWzc**
> >
> > > What happens if you do not have access to the original LLM? (e.g., if you watermarked with ChatGPT but ran verification with Llama-1b, the shift in distributions would be quite large).
> >
> > We have provided a pertinent ablation study in Section F.2, as indicated in Table 9, where we employ a different language model for watermark detection. Although we were unable to conduct experiments directly with ChatGPT, we believe that experiment closely reflects the scenario under consideration by the reviewer.
> >
> > We find that the gamma reweight method is able to substantially maintain watermark strength even when switching models. In the case of the delta reweight scheme, while there is a large decrease in the watermark's detection capability, it still remains functional. As shown in Table 9, the score per token for delta reweight on a different model is about 0.041. This is still significant as, for instance, with 200 tokens, the cumulative score would be approximately 8.2, enabling watermark detection with a p-value of less than 0.0003.
> >
> > Moreover, we have also developed watermark detection methods that are fully independent of the language model. These are detailed in Sections D and F.3 of our appendix. These likelihood-agnostic watermarking methods offer additional flexibility in scenarios where access to the language model is not available.
> >
> > > Would the maximin LLR scheme outperform a z-score based statistical test?
> >
> > To clarify, while both the maximin LLR scheme and z-score scheme are statistical tests and they share the same null hypothesis H0, they test under different alternative hypotheses H1 which corresponds to different watermark distributions. Our maximin LLR scheme aims to distinguish between an unbiased watermark distribution and the original distribution, while the z-score based statistical test is designed to differentiate between a biased watermark distribution and the original distribution.
> >
> > Therefore, a direct comparison between the two must be approached with caution. If a comparison is necessary, one could refer to the AUCs presented in Section F.5 of our paper.
> >
> > > Even for the same model, if you have edits (e.g., you don't have access to the prompt, or if someone has shuffled the sentences in the text), does the maximin LLR scheme work?
> >
> > We have conducted relevant ablation studies concerning these situations. In Section F.2 Table 8 of our appendix, we considered the scenario where one does not have access to the prompt. Furthermore, the impact of text perturbation was explored in Section F.5.
> >
> > ---
> >
> > Once again, we appreciate the reviewer's comments and the opportunity to clarify our results. We noticed that many of the concerns pertain to content in the appendix. We apologize for any inconvenience caused by its omission in the main paper due to space limitations. We encourage the reviewer to consider the information in the appendix for additional context.
> >
> > In closing, we would like to underscore the originality and significant contributions of our work. The introduction of unbiased watermarking methods offer a novel framework that effectively decouples watermark design and quality of language model. This innovation is crucial, as it guarantees the preservation of text quality, and ensures that watermarks remain undetectable to users. We are confident in the unique value and perspective our work brings to the field, and we look forward to its potential impact on future research and applications in language model watermarking.
> >
> > Please let us know if you have any additional questions or concerns.

---

> ### Comment · Reviewer_tWzc · 2023-11-22
> **Typo in my original comment**
>
> I made a typo in my original comment. I mentioned the gamma scheme where I intended to call out the delta scheme (the stronger of the two schemes) is indeed based on inverse transform sampling, as all of the schemes in the cited work. Is this incorrect?

---

> > ### Comment · Reviewer_tWzc · 2023-11-22
> > **Response to rebuttal**
> >
> > 1. I apologise for my earlier typo. I intend to call out the 'gamma' scheme, which the paper itself mentions is based on inverse transform sampling, as are all of the schemes in the Kuditipudi paper. Can the authors clarify why they think the inverse transform sampling based schemes in Kuditipudi et al. paper are very different?
> >
> > 2. Reg AUC as a metric: This goes back to my comment to evaluate with 'instruction-tuned models' with instructions. If I understood the theory correctly, the theory only guarantees that the quality is unaffected but says nothing about the watermarking strength. For the instruction-tuned models, the AUCs are generally not in the ballpark of 1.0. See the Kuditipudi et al. paper for the instruction model experiments, and notice the degradation in the watermark detection performance on moving from Llama to an instruction tuned model. I find the authors claim here in the rebuttal for not running evaluations with instructions on an instruction tuned model confusing -- I would be extremely surprised if the AUC was nearing 1.0 for an Alpaca/Vicuna run at reasonable temperatures.
> >
> > 3. For the experiments in Section F2, what was the parameter d? Can you also look at the robustness of likelihood agnostic methods?
> >
> > 4. " it's nearly impossible to reaching saturation" -- that is besides the actual point. 1. You'll probably never encounter all possible trigrams. E.g., "me apple blue" is probably never going to occur. 2. And some trigrams are quite common, e.g., "This is how". For these common trigrams, you'll stop watermarking after the first time you see it, and you'll gradually weaken the watermark. From my experiment and ballpark numbers, only 70-80% of the trigrams in a 500 word piece of text are unique, and this number drops quite sharply. So, even for that 500 word piece of text, you're not watermarking 20-30% of the tokens... and if you think about 10000 tokens, you'll most definitely not be watermarking more than half the tokens with n=3. Am I missing something here?
> >
> > 5. "The downstream-invariant property is also observed in these new tasks." -- as I said above, my comment had nothing to do with the model quality being affected. It was more to do with the watermarking strength being affected. If your prompt is, "Generate the first 100 primes", do you expect it to be watermarkable? It is known that RL reduces entropy -- do you expect RLHFed model to have the same entropy? Does your theory have any guarantees wrt watermarking strength?

---

> ### Author Response · Authors · 2023-11-23
> **Response to Reviewer tWzc**
>
> Thank you for your continued engagement and for providing detailed feedback. We appreciate the opportunity to address your points.
>
> > I intend to call out the 'gamma' scheme, which the paper itself mentions is based on inverse transform sampling
>
> Is that another typo? Are you referring to the 'delta' scheme?
>
> > Can the authors clarify why they think the inverse transform sampling based schemes in Kuditipudi et al. paper are very different?
>
> In our previous response, we clarified the distinguishing factors between our gamma reweight and inverse sampling. Here, we'll highlight the differences between our delta reweight and Kuditipudi et al.'s inverse transform sampling schemes. Although delta reweight, as a specific case of our framework, employs inverse transform sampling, it diverges significantly in other aspects, such as avoiding watermarking for existing context and deriving score based on LLR and maximin LLR. Additionally, our framework is more general than Kuditipudi et al., as our framework encompasses methods beyond inverse transform sampling, such as the gamma reweight.
>
> > If I understood the theory correctly, the theory only guarantees that the quality is unaffected but says nothing about the watermarking strength.
>
> Regarding our general unbiased watermark framework, you are correct that the framework itself only guarantees that the quality is unaffected but does not specify a lower bound for watermark strength. To clarify, the general framework encompasses a broad range of possibilities, including scenarios where no watermarking is applied at all (with identity map as unbiased reweighting function), the lower bound for watermark strength in this general context is indeed zero.
>
> However, for specific reweights within our framework, there are non-trivial lower bounds on watermark strength. For instance, for delta reweight , the average LLR score for a token with a watermark is its entropy, $H(P)$, by definition. For our likelihood-agnostic methods, the average score is related to the Rényi entropy of order 2, expressed as $\ln2 - \exp(-H_2(P))$. Further details are provided in Appendix D.1.2 of our paper.
>
> > I find the authors claim here in the rebuttal for not running evaluations with instructions on an instruction tuned model confusing -- I would be extremely surprised if the AUC was nearing 1.0 for an Alpaca/Vicuna run at reasonable temperatures.
>
> Per your request, we conducted experiments to provide score per token values for instruction-tuned models:
>
> |Method|Flan (Llama-2-7b-chat-hf)|
> |---|---|
> |$\delta$-reweight|0.2435±0.8345|
> |$\gamma$-reweight|0.0929±0.2841|
>
> These numbers remain substantial, indicating strong watermark detectability.
>
> Moreover, It's crucial to consider the impact of text length on watermark detection. For example, if Llama 2 assists in writing an email with around 100 tokens, the total score would be high, making it easy to distinguish watermarked from unwatermarked text. However, for a short text like a subject line with only about 5 tokens, the discrimination power would be much weaker. Thus, it is essential to carefully consider the context and length of the text when comparing watermark detection capabilities.
>
> > For the experiments in Section F2, what was the parameter d?
>
> The parameter $d$ was chosen from the set $\\{0,0.1,\dots,0.9,1.0\\}$, as detailed in Appendix E, "Watermark detection" section of our paper.
>
> > Can you also look at the robustness of likelihood agnostic methods?
>
> It's important to note that likelihood agnostic methods are independent of the likelihood, so they are invariant to all the perturbations considered in Section F.2. Therefore, their robustness only depends on how much the attacks disrupt the original n-grams in the text.
>
> > you'll stop watermarking after the first time you see it. you'll gradually weaken the watermark ... Am I missing something here?
>
> We think what's missing from your comment is the Pareto optimality of our approach regarding to number of watermarked token and undetectable property.
>
> We would like to reiterate that skipping watermarking is essential to ensure an unbiased watermark, thus preserving the output quality and undetectability.
>
> For example, consider an instruction like "generate a sequence of random 01 string", and assume that the original language model generates perfect binary distribution. After saturation, i.e. after encountering all $2^n$ number of distinct n-grams, we must stop watermarking to avoid making the watermark detectable through n-gram frequency analysis.
>
> We also would like to reiterate that there are straightforward modifications that can discard the $\infty$-shot-undetectable property, and turn to weaker $n$-shot-undetectable property, in exchange for more watermarking opportunities. This is discussed in Section G.2.

---

> ### Author Response · Authors · 2023-11-23
> **Response to Reviewer tWzc**
>
> > If your prompt is, "Generate the first 100 primes", do you expect it to be watermarkable?
>
> No, it's not watermarkable because the entropy is close to zero. Unbiased watermarking is only feasible with enough entropy, as discussed in Appendix G.1.
>
> > It is known that RL reduces entropy -- do you expect RLHFed model to have the same entropy?
>
> We humbly refrain from discussing the impact of RLHF on the entropy of language models, as it is beyond the scope of this paper and our research expertise. However, it is clear that our unbiased watermarking relies on the entropy of language models. Lower entropy makes watermarking more challenging, both in terms of insertion and detection.
>
> > Does your theory have any guarantees wrt watermarking strength?
>
> Yes. For delta reweight , the average LLR score for a token with a watermark is its entropy, $H(P)$, by definition. For our likelihood-agnostic methods, the average score is related to the Rényi entropy of order 2, expressed as $\ln2 - \exp(-H_2(P))$. Further details are provided in Appendix D.1.2 of our paper.

---

> > ### Comment · Reviewer_tWzc · 2023-11-23
> > **Follow up**
> >
> > >>"No, it's not watermarkable because the entropy is close to zero. Unbiased watermarking is only feasible with enough entropy, as discussed in Appendix G.1."
> >
> > Going back to your first comment, do you expect the entropy to be "independent of specific task" or model agnostic? Do you think it's worthwhile understanding well in what situations the proposed scheme works well, and when it fails. Or is that beyond the scope of the paper?
> >
> > >>"In our previous response, we clarified the distinguishing factors between our gamma reweight and inverse sampling. Here, we'll highlight the differences between our delta reweight and Kuditipudi et al.'s inverse transform sampling schemes. Although delta reweight, as a specific case of our framework, employs inverse transform sampling, it diverges significantly in other aspects, such as avoiding watermarking for existing context and deriving score based on LLR and maximin LLR. Additionally, our framework is more general than Kuditipudi et al., as our framework encompasses methods beyond inverse transform sampling, such as the gamma reweight."
> >
> > If I understand correctly here, you do agree that there is significant overlap in the encoding beyond how the keys are generated. That was the point I was trying to make. I highlighted the LLR/Maximin LLR as useful novel contributions, but if you want me to consider them as part of the "delta reweight" scheme, sure. Then I will merge those bullet points.
> >
> > >>Thanks for running this. For Flan Llama, it doesn't seem like the AUC will be near 1.0 at a "few" tokens as claimed in your first response, even with the LLR scheme. Particularly, when you don't have access to the prompt.
> >
> > >>"We humbly refrain from discussing the impact of RLHF on the entropy of language models, as it is beyond the scope of this paper and our research expertise."
> >
> > As you said, the watermarking performance depends on the entropy. I think this understanding is crucial for evaluating whether the scheme proposed here can have a meaningful impact if deployed in the real world. Do the authors think that this is not of interest to their work?
> >
> > If I understood correctly, your theory gives guarantees given underlying entropy but not about how much entropy actually is there. Hence my suggestion to capture an interesting set of tasks in your evaluations.
> >
> > >> Another question: If you're OpenAI, and watermark ChatGPT with your scheme, do you expect them to hand the verification service users their tokenizer for verification as well? I imagine it's a part of their secret sauce, and assuming access to the tokenizer is a strong assumption. This isn't really replicated in the experiments run with changing temperature and dropping a few words.
> >
> > >> "We find that the gamma reweight method is able to substantially maintain watermark strength" is this true even when switching models (e.g., changing tokenizers, and not just changing temperature)?
> >
> > >> "a direct comparison between the two must be approached with caution." -- yes, they're testing different hypothesis but one of the tests uses a lot more information, and might perhaps be impractical in a lot of situations. It's good to understand the trade-offs since you won't be using both if you run a detection service in the real world, or are you saying you would somehow use both to make your decisions?

---

### Meta-Review · Area_Chair_ocE4 · 2023-12-06

**Metareview:**

The paper presents a novel and potentially impactful approach to watermarking LLMs without compromising output quality. The reviewers generally agree that the proposed methods are sound, well-presented, and contribute significantly to the field. However, some concerns are raised regarding the lack of comprehensive evaluations and potential limitations of the approach in practice.

Strengths:

- Concept of "unbiased watermark": Relevant for practical implementation and accountability.
- Novel and effective methodologies: Simple and empirically effective, with theoretical guarantees for unbiasedness.
- New LLR score: Robust and theoretically sound.
- Comprehensive framework: Provides flexibility for design and detection of unbiased watermarks.
- Mathematically well-defined framework: Enables trade-off analysis.

Weaknesses:
- Limited adversarial evaluation.
- Potential limitations: Technical and logistical challenges, impact on performance.
- Complexity of implementation: Requires expertise and resources.

**Justification For Why Not Higher Score:**

The paper has limited adversarial evaluation and the complexity of implementation could be a blocker to get widely used.

**Justification For Why Not Lower Score:**

The reviewers generally agree that the proposed methods are sound, well-presented, and contribute significantly to the field.

---

### Decision · Program_Chairs · 2024-01-16

Accept (spotlight)